# Latent Diffusion for Neural Spiking Data

**Jaivardhan Kapoor**[1][*]     **Auguste Schulz**[1][*]     **Julius Vetter**[1]

**Felix Pei**[1]     **Richard Gao**[1][†]     **Jakob H. Macke**[1,2][†]

[1]Machine Learning in Science, University of Tübingen & Tübingen AI Center, Tübingen, Germany
[2]Department Empirical Inference, Max Planck Institute for Intelligent Systems, Tübingen, Germany
[*]Equal contribution, order determined by a coin toss.
[†]Equal supervision.
`{firstname.lastname@uni-tuebingen.de}`

## Abstract

Modern datasets in neuroscience enable unprecedented inquiries into the relationship between complex behaviors and the activity of many simultaneously recorded neurons. While latent variable models can successfully extract low-dimensional embeddings from such recordings, using them to generate realistic spiking data, especially in a behavior-dependent manner, still poses a challenge. Here, we present Latent Diffusion for Neural Spiking data (LDNS), a diffusion-based generative model with a low-dimensional latent space: LDNS employs an autoencoder with structured state-space (S4) layers to project discrete high-dimensional spiking data into continuous time-aligned latents. On these inferred latents, we train expressive (conditional) diffusion models, enabling us to sample neural activity with realistic single-neuron and population spiking statistics. We validate LDNS on synthetic data, accurately recovering latent structure, firing rates, and spiking statistics. Next, we demonstrate its flexibility by generating variable-length data that mimics human cortical activity during attempted speech. We show how to equip LDNS with an expressive observation model that accounts for single-neuron dynamics not mediated by the latent state, further increasing the realism of generated samples. Finally, conditional LDNS trained on motor cortical activity during diverse reaching behaviors can generate realistic spiking data given reach direction or unseen reach trajectories. In summary, LDNS simultaneously enables inference of low-dimensional latents and realistic conditional generation of neural spiking datasets, opening up further possibilities for simulating experimentally testable hypotheses.

## 1 Introduction

Modern datasets in neuroscience are becoming increasingly high-dimensional with fast-paced innovations in measurement technology [1, 48, 23], granting access to hundreds to thousands of simultaneously recorded neurons. At the same time, the types of animal behaviors and sensory stimuli under investigation have become more naturalistic and complex, resulting in experimental setups with heterogeneous trials of varying length, or lacking trial structure altogether [35, 33, 57]. Therefore, a key target in systems neuroscience has shifted towards understanding the relationship between high-dimensional neural activity and complex behaviors.

For high-dimensional neural recordings, analyses that infer low-dimensional structures have been very useful for making sense of such data [11]. For example, latent variable models (LVMs) are often used to identify neural population dynamics not apparent at the level of single neurons [61, 32, 40]. More recently, deep learning-based approaches based on variational autoencoders (VAEs) [26, 44,

51, 36, 63, 22, 6, 16] have become particularly popular for inferring latent neural representations due to their expressiveness and ability to scale to large, heterogeneous neural recordings with behavioral covariates [36, 63, 16].

However, in addition to learning latent representations, another important consideration is the ability to act as faithful generative models of the data. In other words, models should be able to produce diverse, realistic samples of the neural activity they were trained on, ideally in a behavior- or stimulus-dependent manner. Models with such capabilities not only afford better interpretability analyses and diagnoses for whether structures underlying the data are accurately learned, but have a variety of downstream applications surrounding the design of closed-loop *in silico* experiments. For example, with faithful generative models, one can simulate population responses to hypothetical sensory, electrical, or optogenetic stimuli, as well as possible neural activity underlying hypothetical movement patterns. Most VAE-based approaches focus on the interpretability of the inferred latents, but not the ability to generate realistic and diverse samples when conditioning on external covariates, while sample-realistic models (e.g., based on generative adversarial networks (GANs) [17]) do not provide access to underlying low-dimensional representations. As such, there is a need for models of neural population spiking activity that both provide low-dimensional latent representations *and* can (conditionally) generate realistic neural activity.

Here, we propose **Latent Diffusion for Neural Spiking data (LDNS)**, which combines the ability of autoencoders to extract low-dimensional representations of discrete neural population activity, with the ability of (conditional) denoising diffusion probabilistic models (or, diffusion models) to generate realistic neural spiking data by modeling the inferred low-dimensional continuous representations.

Diffusion models [49, 20, 50] have been highly successful for conditional and unconditional data generation in several domains, including images [20], molecules [59], and audio spectrograms [27] and have demonstrated sampling-fidelity that outperforms that of VAEs and GANs [20]. A key strength of diffusion models that makes them particularly attractive in the context of modeling neural datasets is the ability to flexibly condition the generation on various (potentially complex) covariates, such as to simulate neural activity given certain behaviors. Recently, diffusion models have been extended to continuous neural time series such as local field potentials (LFPs) and electroencephalography

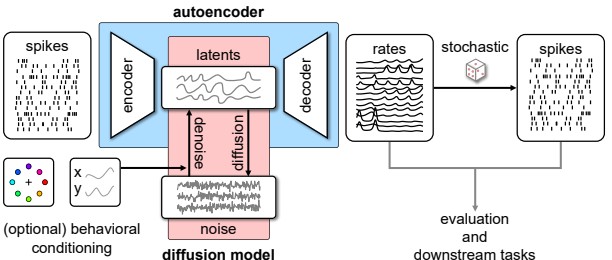

Figure 1: **Latent Diffusion for Neural Spiking data.** LDNS allows for (un)conditional generation of neural spiking data through combining a regularized autoencoder with diffusion models that act on the low-dimensional latent time series underlying neural population activity.

(EEG) recordings [53]. However, due to the discrete nature of spiking data, standard diffusion models cannot be easily applied, thus excluding their use on many datasets in systems neuroscience.

To bypass these limitations, LDNS employs a regularized autoencoder using structured state-space (S4) layers [18] to project the high-dimensional discrete spiking data into smooth, low-dimensional latents without making assumptions about the trial structure. We then train a diffusion model with S4 layers as a generative model of the inferred latents—akin to latent diffusion for images [45], where generation can be flexibly conditioned on behavioral covariates or task conditions.

A fundamental assumption of most low-dimensional latent variable models is that all statistical dependencies between observations are mediated by the latent space. However, in neural spiking data, there are prominent statistical dependencies that ought to persist *conditional* on the latent state, e.g., single-neuron dynamics such as refractory periods, burstiness, firing rate adaptation, or potential direct synaptic interactions. We show how such additional structure can be accounted for in LDNS, by equipping it with an expressive observation model [41, 32, 54, 62]: We use a Poisson model for spike generation with autoregressive couplings which are optimized post hoc to capture the temporal structure of single-neuron activity. This allows LDNS to capture a wide range of biological neural dynamics [55], with only a small additional computational cost.

**Main contributions**   In summary, LDNS is a flexible method that allows for both high-fidelity diffusion-based sampling of neural population activity and access to time-aligned low-dimensional representations, which we validate on a synthetic dataset. Next, we show the utility and flexibility of this approach on complex real datasets: First, LDNS can handle variable-length spiking recordings from the human cortex. Second, LDNS can unconditionally generate faithful neural spiking activity recorded from monkeys performing a reach task. We demonstrate how LDNS can be equipped with an expressive autoregressive observation model that accounts for additional dependencies between data points (e.g., single neuron dynamics), increasing the realism of generated samples. Third, LDNS can generate realistic neural activity while conditioning on either reach direction or full reach trajectories (time series), including unseen behaviors that are then accurately decoded from the simulated neural data. Overall, LDNS enables simultaneous inference of low-dimensional latent representations for single-trial data interpretation and high-fidelity diffusion-based (conditional) generation of diverse neural spiking datasets, which will allow for closed-loop *in silico* experiments and hypothesis testing.

## 2   Methods

### 2.1   Latent Diffusion for Neural Spiking Data (LDNS)

We consider a dataset recorded from a population of $n$ neurons, consisting of trials with spiking data $\mathbf{s} \in \mathbb{N}_0^{n \times T}$ (sorted into bins of fixed length resulting in spike counts over time), and optional simultaneously recorded behavioral covariates $\mathbf{y} \in \mathbb{R}^n$ (that can also be time-varying $\mathbf{y} \in \mathbb{R}^{n \times T}$). A dataset of $M$ such trials $\mathcal{D}$ can be written as $\mathcal{D} = \{\mathbf{s}^{(i)}, \mathbf{y}^{(i)}\}$, possibly with varying trial lengths $T_1 \ldots T_M$. We make the assumption that a large fraction of the variability in this dataset can be captured with a few underlying latent variables $\mathbf{z} \in \mathbb{R}^{d \times T}$, where $d < n$.

Our goal is to generate realistic spiking data $\mathbf{s}^*$ that faithfully capture both population-level and single-neuron dynamics of $\mathbf{s}_{1 \ldots T}$ with the ability to optionally condition the generation on behavior $\mathbf{y}_{\text{cond}}$. To this end, we propose a new method, LDNS, that combines the strength of neural dimensionality reduction approaches with that of diffusion-based generation.

LDNS uses a two-stage training framework, adopted from the highly successful family of latent diffusion models (LDMs) [45, 8, 59]. To train LDNS, we first train a regularized autoencoder [14] to compress the spiking data into a low-dimensional continuous latent space (Fig. 1). Concretely, we focus on two objects of interest for the LDNS autoencoder: **(1)** inferring a time-aligned, low-dimensional smooth representation $\mathbf{z} \in \mathbb{R}^{d \times T}$ that preserves the shared variability of the spiking data, and **(2)** predicting smooth firing rates $\lambda$ that are most likely to give rise to the observed spiking data.

In the second stage, we train a diffusion model in latent space, possibly employing *conditioning* to make generation contingent on external (e.g., behavioral) covariates (Fig. 1). For the diffusion model, our main objective is the generation of $\mathbf{z}^* \in \mathbb{R}^{d \times T}$ that captures the distribution of inferred autoencoder latents. We also want the ability to sample latent trajectories of varying length.

In both stages, we use structured state-space (S4) [18] layers for modeling temporal dependencies. S4 layers consist of state-space transition matrices that can be unrolled into arbitrary-length convolution kernels, allowing sequence modeling of varying lengths. For details on network architectures and S4 layers, see appendix A1.

### 2.2   Regularized autoencoder for neural spiking data

For the spiking data, we choose a Poisson observation model, and train autoencoders by minimizing the Poisson negative log-likelihood of the input spikes $\mathbf{s}$ given the predicted rates $\lambda = \texttt{decoder}(\mathbf{z})$. To enforce smoothness in the latent space, where $\mathbf{z} = \texttt{encoder}(\mathbf{s})$, we add an $L_2$ regularization along with a temporal smoothness regularizer over $\mathbf{z}$, resulting in the combined loss

$$\mathcal{L}_{\text{AE}} = \mathbb{E}_{\mathbf{s} \sim \mathcal{D}} \left[ \underbrace{\sum_{\substack{i=1 \\ t=1}}^{n,T} (\lambda_i(t) - s_i(t) \ln \lambda_i(t))}_{\text{Poisson NLL}} + \beta_1 \underbrace{\|\mathbf{z}\|^2}_{L_2 \text{ reg.}} + \beta_2 \sum_{\substack{k=1 \\ t=k+1}}^{K,T} \underbrace{\frac{\|\mathbf{z}(t) - \mathbf{z}(t-k)\|^2}{(1+k)}}_{\text{temporal smoothness}} \right]. \quad (1)$$

---

Code available at https://github.com/mackelab/LDNS.

To prevent the autoencoder from predicting highly localized Poisson rates, which have sharp peaks at input spike locations, we further regularize training using coordinated dropout [24], i.e., we randomly mask input spikes and compute the loss on the predicted rates at the masked locations (details in appendix A1.2).

**Accounting for single-neuron dynamics with an expressive observation model**    So far, LDNS (like most latent variable models for neural data) uses a Poisson observation model, which assumes that all statistical dependencies are mediated by the latent state. To address this limitation and to capture dynamics and variability, which are "private" to individual neurons (such as refractory periods or burstiness), we propose to learn an autoregressive observation model. We make the predicted Poisson rates for each neuron $i$ dependent also on recent spiking history, by including additional spike history couplings $h_i$ [41, 32], resulting in the observation model

$$s_i(t) \sim \exp\left(\log \lambda_i(t) + h_{i,0} + \sum_{\tau=1}^{T'} h_{i,\tau} s_i(t-\tau)\right), \qquad (2)$$

where $T'$ corresponds to the time-lagged window length. This modification is learned post hoc, and the parameters $h_i$ are fit with a maximum-likelihood objective (details in appendix A1.3). This approach does not alter the latent dynamics, while augmenting the model with single-neuron autoregressive dynamics. We observe that including spike history increases the realism of generated data and enables us to accurately capture single-neuron autocorrelation structures (Sec. 3.4).

### 2.3   Denoising Diffusion Probabilistic Models

In the second stage of training, we train diffusion models [20] to generate (conditional) samples from the distribution of inferred latents. The training dataset therefore contains autoencoder-derived latents of each trial, and optionally, additional conditioning information such as the corresponding behavior, i.e., $\mathcal{D}_z = \{\mathbf{z}^{(\mathbf{i})} = \texttt{encoder}(\mathbf{s}^{(i)}), \mathbf{y}^{(i)}\}$.

Diffusion models aim to approximate the data distribution $q(\mathbf{z})$ through an iterative denoising process starting from standard Gaussian noise. For latent $\mathbf{z}$ (denoted as $\mathbf{z}_0$ for diffusion timestep 0), we first produce a noised version at step $t$ by adding Gaussian noise of the form $q(\mathbf{z}_t|\mathbf{z}_0) = \mathcal{N}\left(\sqrt{\bar{\alpha}_t}\mathbf{z}_0, (1-\bar{\alpha}_t)I\right)$. Here, $\bar{\alpha}_t = \prod_{k=1}^t \alpha_k$, where the noise scaling factors $\alpha_1 \ldots \alpha_T$ follow a fixed linear schedule. We then train a neural network to approximate the reverse process $p_\theta(\mathbf{z}_{t-1}|\mathbf{z}_t)$ for each diffusion timestep. The true (denoising) reverse transition $q(\mathbf{z}_{t-1}|\mathbf{z}_t)$ is intractable—however, we can apply variational inference to learn the *conditional* reverse transition $q(\mathbf{z}_{t-1}|\mathbf{z}_t, \mathbf{z}_0)$, which has a closed form written as

$$q(\mathbf{z}_{t-1}|\mathbf{z}_t, \mathbf{z}_0) = \mathcal{N}\left(\frac{\sqrt{\alpha_t}(1-\bar{\alpha}_{t-1})}{1-\bar{\alpha}_t}\mathbf{z}_t + \frac{\sqrt{\bar{\alpha}_{t-1}}(1-\alpha_t)}{1-\bar{\alpha}_t}\mathbf{z}_0, \frac{(1-\alpha_t)(1-\bar{\alpha}_{t-1})}{1-\bar{\alpha}_t}I\right). \qquad (3)$$

We train the neural network $\mu_\theta(\mathbf{z}_t, t)$ to approximate the mean of this distribution by optimizing the loss $\mathbb{E}_{\mathbf{z}_0 \sim \mathcal{D}_z, \epsilon_0, t}\|\epsilon_\theta(\mathbf{z}_t, t) - \epsilon_0\|^2$, where $\epsilon_0$ is the noise used to generate $\mathbf{z}_t$ from $\mathbf{z}_0$, and $\epsilon_\theta(\mathbf{z}_t, t)$ is the equivalent reparameterization for $\mu_\theta(\mathbf{z}_t, t)$. At test time, we sequentially sample $\mathbf{z}_{t-1}$ given $\mathbf{z}_t$ using the learned transition $p_\theta(\mathbf{z}_{t-1}|\mathbf{z}_t)$, starting from standard Gaussian noise. Using S4 layers in the denoising network allows us to generate latents with varying lengths. This is achieved by unrolling the state transition matrix in the S4 layers to the desired length for each denoising step.

Diffusion models may be conditioned on fixed-length and time-varying covariates $\mathbf{y}$, in which case we learn the approximate reverse transition $p_\theta(\mathbf{z}_{t-1}|\mathbf{z}_t, \mathbf{y})$. Details on the conditioning mechanisms in appendix A1.4.

## 3   Experiments and Results

### 3.1   Datasets and tasks

We first evaluate the performance of LDNS on a synthetic spiking dataset where we have access to the ground-truth firing rates and latents. We choose the Lorenz attractor [29] as a low-dimensional, non-linear dynamical system commonly used in neuroscience [7, 36]. We simulate rates as an affine mapping from the 3-dimensional system to a 128-dimensional neural space, and sample from a

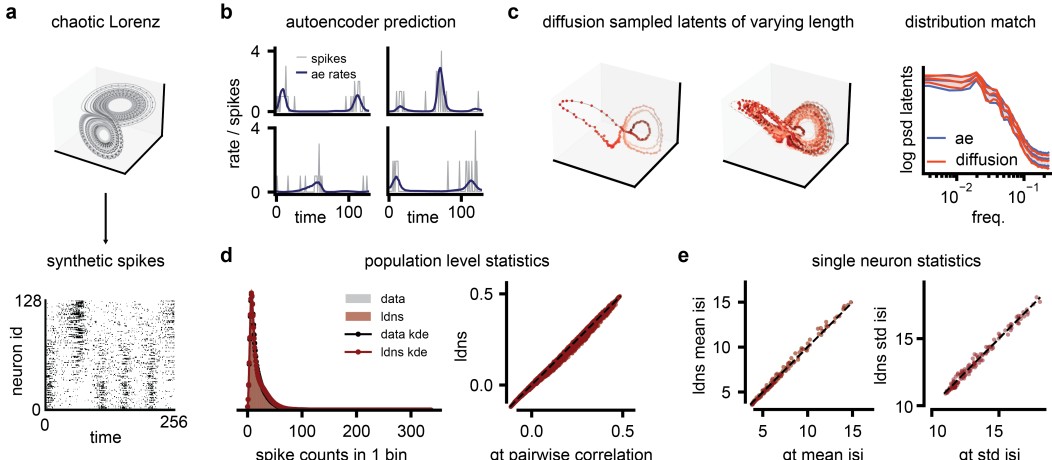

Figure 2: **Realistic generation of spiking data with underlying chaotic dynamics. a)** Synthetic spiking data from an underlying Lorenz system with a Poisson observation model. **b)** Accurate, smooth rate predictions of the autoencoder for held-out spiking data. **c)** Plotted trace of sampled latents (256 bins training length, left) and $16\times$ the original training length (middle). The sampled latent distribution matches the PSD of the autoencoder latents (right; median, 10%, and 90% percentiles). **d)** LDNS population spike count histogram (kde: kernel density estimate) and pairwise cross-correlations match the training distribution. **e)** LDNS single neuron statistics, i.e., mean inter-spike interval (isi) and std isi, match the training distribution.

Poisson distribution to generate spiking data. Next, we showcase the applicability of LDNS on two neural datasets: We apply our method on a highly complex dataset of human neural activity (128 units) recorded during attempted speech [56]. This dataset poses a challenge to many modeling approaches due to the different imagined sentences, resulting in variable lengths of the neural time series (between 2-10 seconds with a sampling rate of 50 Hz). Finally, we apply LDNS to model premotor cortical activity (182 units) recorded from monkeys performing a delayed center-out reach task with barriers [9, 38]. The multi-modal nature of the dataset allows us to assess both unconditional as well as conditional generation of neural spiking activity given monkey reach directions and entire velocity profiles of the performed reaches. See appendix A2,A3 for data and training details.

For the unconditional generation of monkey reach recordings (Sec. 3.4), we train both a Poisson observation model as well as a spike history-dependent autoregressive observation model. For all other experiments, we only train a Poisson observation model.

**Baselines**  We compare LDNS to the most commonly known VAE-based latent variable model: Latent Factor Analysis via Dynamical Systems (LFADS [51, 36, 47]), which has been shown to outperform various classical latent variable models on a variety of tasks ([38], details in appendix A4). To ensure that we use optimal hyperparameters for LFADS, we follow the auto-ML pipeline proposed by Keshtkaran et al. [25]. This approach, termed AutoLFADS, has been shown to perform better than the original LFADS on benchmark tasks [38]. For the unconditional generation of monkey reach recordings, we further compared to additional VAE baselines [21, 62] (appendix A5).

**Metrics**  For all experiments, we assess how well LDNS-generated samples match the spiking data in the training distribution. Concretely, we compare population-level statistics by computing 1) the distribution over the population spike count, which sums up all spikes co-occurring in the population in a single time bin (i.e., spike count histogram), and 2) pairwise correlations of LDNS samples and the spiking data for each pair of neurons. For single-neuron statistics, we compare 3) the mean and 4) standard deviation of the inter-spike-interval distribution for each neuron (mean isi and std isi). When multiple spikes occur in a single time bin, the spike times are distributed equally in this bin [12]. To further evaluate population dynamics, we compare the principal components of smoothed spikes.

### 3.2 LDNS captures the true spiking data distribution with an underlying Lorenz system

We simulate trials of length 256 timesteps from the three-dimensional (chaotic) Lorenz system (Fig. 2a). The regularized autoencoder extracts smooth latent time series (eight latent dimensions)

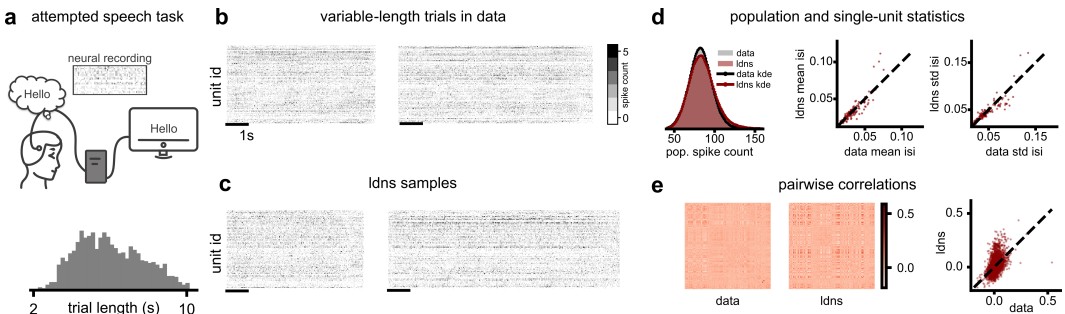

Figure 3: **Unconditional generation of variable-length trials of human spiking data during attempted speech. a)** Multi-unit activity is recorded from speech production-related regions of the brain (top) during attempted vocalization of variable-length sentences (bottom). **b)** Neural activity during sentences of different lengths. **c)** LDNS unconditionally sampled trials with different lengths, using the Poisson observation model. **d)** LDNS population spike count histogram, and mean and std of the isi match those of the data. **e)** Correlation matrices of the data (left) and LDNS samples (middle), and scatterplot of the pairwise correlations of data vs. LDNS samples (right).

from the 128-dimensional spiking data, resulting in smooth firing rate predictions that closely match the ground-truth rates (Fig. 2b, Supp. Fig. A2,A3). We then train a diffusion model on the extracted autoencoder latents. Latents sampled from the diffusion model (red) preserve the attractor geometry of the Lorenz system (Fig. 2c, left, three of the eight latent dimensions), indicating that LDNS preserves a meaningful latent space. The architectural choice of S4 layers allows for length generalization: although we train on time segments of 256-time steps, we can sample and successfully generate latent trajectories that are much longer, but still accurately reflect the Lorenz dynamics (Fig. 2c, middle, $16\times$ longer generation). In comparison, LFADS exhibits instabilities when generating such longer sequences (appendix A6.1). Overall, the latent time series distribution is captured well by the diffusion model, with matching power spectral densities (PSD) per latent dimension (Fig. 2c, right, other dimensions in Supp. Fig. A3).

To assess the sampling fidelity of the generated synthetic neural activity, we compute a variety of spike statistics frequently used in neuroscience. LDNS captures both population-level statistics, such as the population spike count histogram and pairwise correlations between neurons (Fig. 2d), as well as single-neuron statistics, quantified by the mean and standard deviation of inter-spike-intervals (Fig. 2e). LDNS also captures the temporal correlation structure of the data (Supp. Fig. A4). These results demonstrate that LDNS can both perform inference of low-dimensional latents and provide high-fidelity diffusion-based generation that perfectly captures the statistics of the ground-truth synthetic data.

### 3.3 Modeling variable-length trials of neural activity recorded in human cortex

Next, we assess whether LDNS is capable of capturing real electrophysiological data, applying it to neural recordings from human cortex during attempted speech (Fig. 3a, top, Willett et al. [56]). A participant with a degenerative disease who is unable to produce intelligible speech attempts to vocalize sentences prompted on a screen, while neural population activity is recorded from the ventral premotor cortex. Since there is a large variation in the length of prompted sentences (Fig. 3a, bottom), this dataset allows us to evaluate the performance of LDNS on real data in naturalistic settings with variable-length and highly heterogeneous dynamics.

To account for varying trial length during autoencoder training, we pad all trials to a maximum length of 512 bins and compute the reconstruction loss only on the observed time bins. For the diffusion model, we indicate the target trial length with a binary mask as a conditioning variable.

This approach allows us to infer time-aligned latents underlying the cortical activity of the participants, compressing the population activity by a factor of four before training an unconditional diffusion model on these latents. Resulting samples of LDNS, mimicking human cortical activity, are visually indistinguishable from the real data (Fig. 3b,c, additional samples in Supp. Fig. A7). This is reflected in closely matched population spike count histograms (Fig. 3d, left), and single neuron statistics such as mean and standard deviation of the inter-spike interval (Fig. 3d, right). Additionally, real

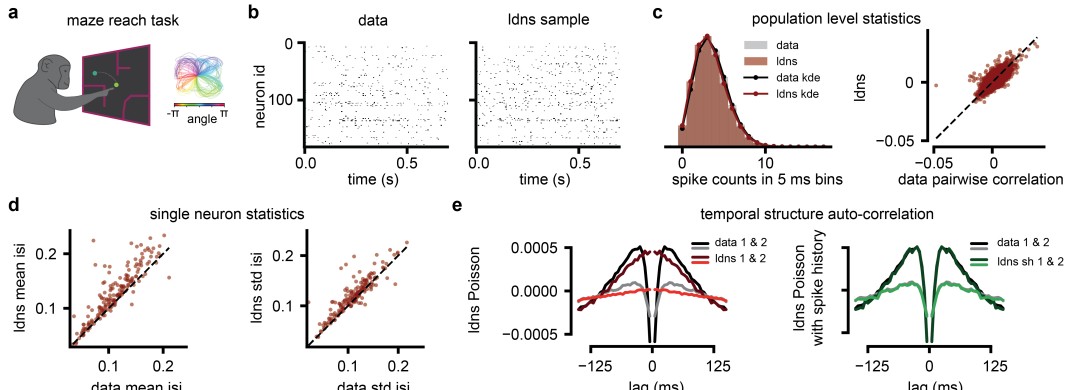

Figure 4: **Realistic generation of spiking data in a monkey performing reach tasks. a)** A monkey performs diverse reach movements in different mazes. **b)** Neural activity during a reach trial and a sampled trial from LDNS with a Poisson observation model. **c)** The LDNS population spike count histogram, and pairwise correlations match those of the data. **d)** LDNS mean- and std isi match the monkey data distribution. **e)** Auto-correlation of data, LDNS samples with Poisson observations (left), and LDNS samples with spike history, grouped according to correlation strength.

and LDNS-sampled spikes display similar population dynamics, as reflected in the top principal components (Supp. Fig. A8). While LDNS tends to overestimate some pairwise correlations, it captures prominent features of the correlation structure in the data (Fig. 3e, Pearson correlation coefficient $r = 0.47$), and our analysis indicates that this slight mismatch already arises at the autoencoder stage (Supp. Fig. A9).

LDNS allows for both inferring latent representations and generating variable-length trial data, making it applicable to complex real neural datasets without a fixed trial structure.

### 3.4 Realistic generation of spiking data from a monkey performing reach tasks

We further evaluate LDNS in a different setting by applying it to model sparse spiking data recorded from a monkey performing a reaching task constrained by barriers that form a maze (Fig. 4a, left). The variety of different maze architectures leads to diverse reach movements of both curved and straight reaches (Fig. 4a, right). We again infer low-dimensional latent trajectories that capture the shared variability of the neural population and then train an unconditional diffusion model on these latents. Sampled spikes from LDNS closely resemble the true, sparse population data (Fig. 4b, additional samples in Supp. Fig. A11), and closely match population-level spike statistics (Fig. 4c). Single neuron statistics in this low spike count regime (a maximum of three spikes per neuron in 5 ms bins) are also captured well (Fig. 4d), and are on par with or better than LFADS [36, 25] (see Table 1 for summary of main comparisons, and appendix A5 for additional baselines [21, 62]). Beyond spiking statistics, we observe that LDNS also preserves the temporal structure of population dynamics, as reflected in the top principal components of smoothed spikes (Supp. Fig. A15). Thus, LDNS can generate spiking data that is faithful at the level of both single-neuron and population dynamics.

Table 1: **Model metrics comparison.** $D_{KL}$ for the population spike count histogram and RMSE comparisons. Mean and standard deviation across 5 folds sampled with replacement. **sh** represents observation models with spike history. **Bolded** entries represent best-performing values for Poisson and spike-history observation models.

| Method | $D_{KL}$ psch | RMSE pairwise corr | RMSE mean isi | RMSE std isi |
|---|---|---|---|---|
| AutoLFADS | $0.0040 \pm 2.2$e-4 | $0.0026 \pm 1.25$e-5 | $0.039 \pm 0.003$ | $0.029 \pm 0.001$ |
| LDNS | $\mathbf{0.0039 \pm 3.9}$**e-4** | $\mathbf{0.0025 \pm 1.1}$**e-5** | $\mathbf{0.037 \pm 0.001}$ | $\mathbf{0.023 \pm 0.001}$ |
| AutoLFADSsh | $0.0036 \pm 2.1$e-4 | $0.0026 \pm 1.8$e-5 | $0.034 \pm 0.002$ | $\mathbf{0.023 \pm 0.0001}$ |
| LDNSsh | $\mathbf{0.0016 \pm 6.2}$**e-4** | $\mathbf{0.0025 \pm 1.07}$**e-5** | $\mathbf{0.024 \pm 0.002}$ | $0.023 \pm 0.001$ |

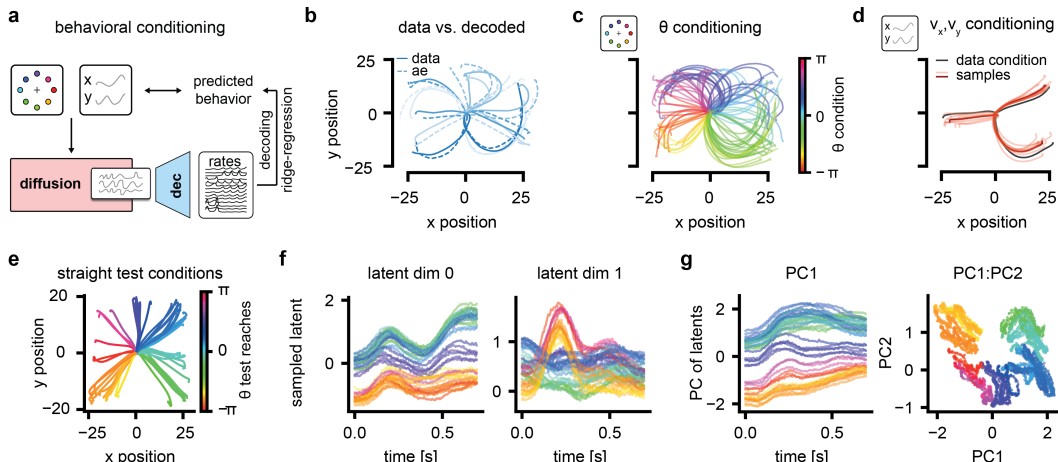

Figure 5: **Generation conditioned on monkey reach directions and velocity traces. a)** Closed loop assessment: do conditionally generated latents translate to neural activity consistent with the desired direction or reach movement? **b)** Unseen reach movements (data) and corresponding movements decoded from the rates predicted by the autoencoder (ae). **c)** Decoded reach directions of LDNS samples conditioned on initial reach angles $\theta$. **d)** Decoded reach directions of LDNS samples conditioned on 3 unseen reach movements (velocities $v_x, v_y$). **e)** Straight reaches from the test set used for velocity conditioning. **f)** LDNS sampled latents conditioned on trajectories shown in e) vary smoothly over time and reflect information about reach angles. **g)** PCs of sampled LDNS latents shown in f) reveal meaningful and separable information about behavior.

Both LFADS and the LDNS autoencoder are optimized by maximizing the Poisson log-likelihood, and thus cannot capture single-neuron dynamics such as refractoriness [41, 55], which can have a strong influence on the observed autocorrelation structure. Given the overall sparsity of the spiking data and resulting low correlations (Supp. Fig. A10), we focus on the temporal structure of auto-correlations averaged within groups of neurons ($\approx 45$ neurons per group) split by their instantaneous correlation strength [32]: darker colors correspond to the highest correlated group of four, lighter colors correspond to the group with the second highest correlations (Fig. 4e). We then compare these auto-correlations to those of grouped LDNS samples with a Poisson observation model (red). As expected, LDNS with Poisson observations is unable to capture the dip in the data auto-correlation at 5 ms lags (one time bin) (Fig. 4e, left).

To overcome this mismatch, we train an additional spike history-dependent autoregressive observation model on top of the inferred rates (LDNSsh, for spike history). In contrast to the Poisson samples, autoregressive samples can capture this aspect of neural spiking data very accurately while also improving the overall fit to the empirical auto-correlation (Fig. 4e, right). Moreover, the post hoc optimization of these filters also improves modeling of other single-neuron, as well as population-level statistics, such as the population spike count histogram or the mean of the isi (Table 1, Supp. Fig. A13).

We view this post-hoc augmentation as a key modular contribution, which can be flexibly applied to other generative models. To this end, we extend AutoLFADS with spike history dependence (LFADSsh), improving its performance across metrics. The augmented LFADSsh also captures the dip in autocorrelation at 5 ms lags (Supp. Fig. A12). Still, in both observation model variants, LDNS maintains superior or comparable performance (Table 1).

Thus, Poisson LDNS allows for the generation of spiking data that is on par or better in terms of sampling fidelity than previous approaches. Incorporating spike-history dependence and sampling spikes autoregressively allows us to further increase the realism of generated spike trains, leading to a large improvement on several of the considered metrics.

### 3.5 Conditional generation of neural activity given reach directions or velocity profiles

Lastly, we assess the ability of conditional LDNS to generate realistic neural activity conditioned on behavioral covariates of varying complexity: the reach angle or entire velocity time series (Fig. 5a). We first validate that the autoencoder predicts firing rates that allow us to linearly decode the behavior

following the ridge-regression approach proposed in [38]. Decoded behavior from autoencoder reconstructed rates matches the true trajectories of unseen test trials (Fig. 5b).

Given that the autoencoder performs adequately, we then test the ability to generate neural time series conditioned on the initial reach angle performed by the monkey $\theta_{\text{reach}}$. Indeed, from the generated samples of neural activity, we can decode—using the same linear decoder—realistic reach kinematics that are consistent with the conditioning angle $\theta_{\text{reach}}$ and overall reach kinematics (Fig. 5c). This indicates that LDNS can generate realistic neural activity consistent with a queried reach direction.

An even more challenging task that is intriguing for hypothesis generation is the ability to mimic an entire experiment and ask what the neural activity *would have looked like* if the monkey had performed a particular hypothetical movement. To this end, we train a diffusion model on the same autoencoder-inferred latents but now condition on entire velocity traces (Fig. 5d). Velocity-conditioned LDNS is able to produce different samples of neural activity that are consistent with, but not exact copies of, the reach trajectories of the held-out trials given as the conditioning covariate. Such closed-loop conditioning experiments open the possibility of making predictions about neural activity during desired unseen behaviors, and thus make experimentally testable predictions.

Finally, to understand how LDNS incorporates behavioral information, we analyzed latent trajectories that were conditionally sampled based on straight reach movements in different directions (Fig. 5e). Individual samples of latent trajectories vary smoothly within a trial (Fig. 5f), while reach direction varies smoothly across samples in the first principal component (PC1) of the latents (Fig. 5g, left). Projection onto the first two PCs of latent trajectories shows clear clustering by reach direction (Fig. 5g, right), and we show that such clustering arises already at the autoencoder stage (Supp. Fig. A17).

In summary, LDNS not only produces faithful spiking samples but also allows for flexible conditioning. Furthermore, LDNS learns an interpretable latent space with behaviorally-relevant structure.

## 4 Related Work

**Latent variable models of neural population dynamics**   LDNS builds on previous LVMs in neuroscience, which have been extensively applied to infer low-dimensional latent representations of neural spiking data [61, 32, 39, 58, 62, 13, 28, 63] (see [38] for a comprehensive list.) In addition to capturing shared population-level dynamics and dependence on external stimuli [4], LVMs have been extended to allow autoregressive neuron-level (non-Poisson) dynamics [32, 13, 62] or even direct neural interactions [54]. While these methods often have useful inductive biases (e.g., linear dynamical systems [32, 28] or Gaussian process priors [61]), these models are typically not expressive enough to yield realistic neural samples across a range of conditions.

**Deep LVMs and other deep learning-based approaches**   Variational autoencoders (VAEs) [26] are particularly popular in neuroscience as they allow us to infer low-dimensional dynamics underlying high-dimensional discrete data [63, 16, 46], especially when combined with nonlinear recurrent neural networks [36, 21]. VAEs have been used to infer identifiable low-dimensional latent representations conditioned on behavior [63, 21] and have incorporated smoothness priors using Gaussian Processes to regularize the latent space [16]. However, the generation performance of VAEs is rarely explored in neuroscience. Besides VAEs, generative adversarial networks (GANs [17]) have been proposed to synthesize spiking neural population activity [34, 42]. While GANs produce high-fidelity samples, they are challenging to train reliably and lack a low-dimensional latent space. More recently, transformer-based architectures have also been adapted to model neural activity [5, 60], though often with the focus of accurate decoding of behavior instead of generation of realistic spiking samples, while also lacking an explicit latent space [3]. Lastly, deterministic approaches utilizing RNNs for dynamical systems reconstruction also target low-dimensional latent dynamics underlying neural data [19], but they do not act as probabilistic generative models.

**Diffusion models**   LDNS leverages recent advances in diffusion models, which have become state-of-the-art for high-fidelity generation in several domains [20, 27], including continuous-valued neural signals such as EEG [53], as well as in time series forecasting and imputation tasks [2, 52, 43]. Similar to the LDNS architecture, Alcaraz and Strodthoff [2] also use an S4-based denoiser for imputation. More specifically, LDNS is inspired by latent diffusion models [45, 27, 59, 15], which benefit from operating on the latent space of an autoencoder and flexible conditioning mechanisms to generate samples based on a given covariate, as is done with text-to-image [45] and other cross-

modality scenarios. Conveniently, this allows LDNS to bypass the challenges of directly modeling discrete-valued spiking data, by instead transforming spikes into the continuous latent space.

## 5   Summary and discussion

We here proposed LDNS, a flexible generative model of neural spiking recordings that simultaneously infers low-dimensional latent representations *and* generates realistic neural activity conditioned on behavioral covariates. We apply LDNS to model three different datasets: synthetic data simulated from chaotic Lorenz dynamics, human cortical recordings with heterogeneous and variable-length trials, and finally, neural recordings in monkeys performing reach actions in a maze. Through our experiments, we demonstrate how several features of LDNS are beneficial for modeling complex datasets in neuroscience:

First, following other LDMs in the literature, LDNS decouples latent inference and probabilistic modeling of the data, offering flexibility in reusing the trained autoencoder and diffusion model. For the monkey recordings, all diffusion models (unconditional, conditioned on reach angle, and conditioned on hand velocities) operate in the latent space of the same autoencoder, in contrast to existing approaches that require end-to-end retraining for each type of conditioning variable. LDNS is also faster to train than AutoLFADS, which requires population-based training to optimize hyperparameters (appendix A3.1). Second, we show that LDNS autoencoders can be augmented with per-neuron autoregressive dynamics to capture single-neuron temporal dynamics (e.g., refractoriness), which otherwise cannot be captured with population-level shared dynamics. Third, as a result of the length-generalizable autoencoders and diffusion models using S4 layers, LDNS can generate variable-length trials in both the Lorenz example and human cortical recordings—a feature that will be particularly useful in modeling datasets recorded during naturalistic stimuli or behavior.

Altogether, these features enable LDNS to generate realistic neural activity, especially when conditioned on behavioral covariates. In our experiments, we demonstrate that unseen movement trajectories can be used to conditionally generate samples of neural activity, from which we can decode these hypothetical behaviors. These generated latent trajectories reflect behavioral information in an interpretable way. Our methodology is general and can be applied to recordings from any brain region, beyond the motor and speech cortex examples shown here. Thus, LDNS opens up further possibilities for hypothesis generation and testing *in silico*, potentially enabling stronger links between experimental and computational works.

**Limitations**   In real neural data, the latent dimensionality of the system is not known, and as with all LVMs (which often assume that population dynamics are intrinsically low-dimensional), choosing an appropriate latent dimension can be challenging. Furthermore, any modeling errors at the encoding and decoding stage of the autoencoder will affect the overall performance of the latent diffusion approach. Nevertheless, in our experiments, we found that autoencoder training is fast, stable, and reasonably robust to hyperparameter configurations. While LDNS was still able to model the data well under relatively severe compression (e.g., 182-to-16 for the monkey recordings), optimizing latent dimensionality to balance expressiveness and interpretability remains a goal for future research.

**Broader impact**   Realistic spike generation capabilities increase the risk of research manipulation by generating synthetic data that may be difficult to detect. On the other hand, LDNS could be useful for the dissemination of privatized clinical data, though we acknowledge the critical importance of protecting data privacy when working with sensitive human participant data. Finally, synthetically generated data (conditioned on unseen behavioral conditions) could be useful for augmenting the training of brain-computer interface decoding models.

## Acknowledgements

This work was supported by the German Research Foundation (DFG) through Germany's Excellence Strategy (EXC-Number 2064/1, PN 390727645) and SFB1233 (PN 276693517), SFB 1089 (PN 227953431), SPP2041 (PN 34721065), the German Federal Ministry of Education and Research (Tübingen AI Center, FKZ: 01IS18039), the Human Frontier Science Program (HFSP), and the European Union (ERC, DeepCoMechTome, 101089288). We utilized the Tübingen Machine Learning Cloud, supported by DFG FKZ INST 37/1057-1 FUGG. JK, AS, and JV are members of the International Max Planck Research School for Intelligent Systems (IMPRS-IS) and JV is supported

by the AI4Med-BW graduate program. We thank Chethan Pandarinath for providing access to their compute cluster to train AutoLFADS. We thank Christian F. Baumgartner and all Mackelab members for feedback and discussions. We would like to also thank our reviewers for their insightful comments which improved our paper.

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

# Appendix

## A1 LDNS architecture

Here we describe the exact network components and architecture for the autoencoder and diffusion model.

### A1.1 Structured State-Space Layers (S4)

A central component of our autoencoder architecture is the recently introduced structured state space models (S4)[18]. With an input sequence $\mathbf{x} = [x_1 \ldots x_T] \in \mathbb{R}^T$ and corresponding output $\mathbf{y} = [y_1 \ldots y_T] \in \mathbb{R}^T$, an S4 layer applies the following operation for each timestep –

$$s_t = \overline{A}s_{t-1} + \overline{B}x_t \tag{4}$$
$$y_t = Cs_t,$$

where the discretized state and input matrices $\overline{A}, \overline{B}$ given continuous analogues $A, B$ and step size $\Delta$ are computed as

$$\overline{A} = (I - \Delta/2 \cdot A)^{-1}(I + \Delta/2 \cdot A) \tag{5}$$
$$\overline{B} = (I - \Delta/2 \cdot A)^{-1}\Delta B.$$

When the state $s_t$ is not required, this recurrent computation of the output $\mathbf{y}$ given input sequence $\mathbf{y}$ can be unrolled into a parallelizable convolution operation

$$\mathbf{y} = K * \mathbf{x}, \text{ with unrolled kernel} \tag{6}$$
$$K = (C\overline{B}, C\overline{AB}, \ldots, C\overline{A}^{T-1}\overline{B}).$$

We used the S4 implementation[1] provided by Gu et al. [18] that stably initializes the state transition matrix $A$ using a diagonal plus low-rank approximation. For a multivariate input-output pair $\mathbf{x}, \mathbf{y} \in \mathbb{R}^{D \times T}$, we apply $D$ separate univariate S4 layers for each dimension and then mix them in the channel-mixing layer using an MLP (see next section). Each univariate input-output mapping consists of $H$ separate S4 "head" that are expanded and contracted from and into a single dimension.

Due to its recurrent nature, S4 is a causal layer, enabling variable-length training and inference. To enable bidirectionality, we flip the input signal $\mathbf{x}$ in time, apply $H/2$ S4 heads each for the flipped and unflipped signal, and then combine these at the end into univariate signal $\mathbf{y}$. This allows bidirectional flow of information from front-to-back and back-to-front of the signal.

### A1.2 Autoencoder

We include temporal information only in the encoder and model the decoder as a lightweight pointwise MLP for the autoencoder (Supp. Fig. A1). This allows us to temporally align the latents with the signal, and ensure that no further temporal dynamics are introduced when mapping the latents back into Poisson rates.

We use causal S4 layers, allowing length generalization and handling of variable-length signals. During training, we pad the input spiking data with zeros into a fixed length and only backpropagate through the unpadded output rates.

Furthermore, to infer smooth rates and avoid spiking behavior, we use coordinated dropout [24]. For each time bin independently, we mask the input spikes to zero with random probability $p$ and scale up the remaining spikes by $\frac{1}{1-p}$ (this preserves the firing statistics of the spiking data). We then backpropagate through the Poisson NLL loss only over the masked positions, effectively preventing the network from collapsing to a spiking prediction of the Poisson rates.

---

[1]Github link: https://github.com/state-spaces/s4/blob/main/models/s4/s4.py

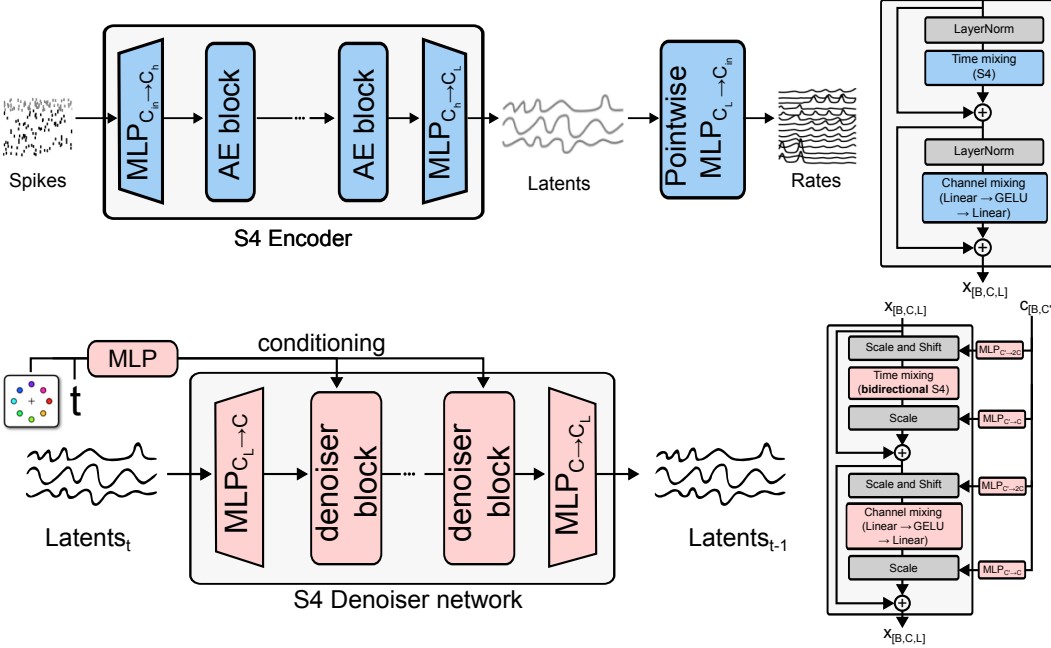

Figure A1: (Top left) The S4 autoencoder architecture. (Top right) Architecture for the autoencoder blocks used in the encoder. (Bottom left) The S4 diffusion model architecture. (Bottom right) Architecture for the diffusion blocks.

### A1.3 Spike-history-augmented Poisson model

The parameters of the autoregressive observation model in Eq. (2) are learned by maximizing the Poisson log-likelihood. Training is performed jointly for all neurons with a history length of $T' = 20$, corresponding to 100 ms, using the AdamW optimizer [30] (learning rate 0.1, weight decay 0.01). In our implementation, we use the Softplus function given by $\mathrm{Softplus}(x) = \log(1 + \exp(x))$ as an approximation to the exponential function in Eq. (2), which is accurate for the low-count regime while increasing numerical stability. During autoregressive sampling, we limit the maximum possible spike count to 5 spikes, which corresponds to the biological maximum, limited by the refractory period for 5 ms time bins.

### A1.4 Diffusion model

We consider four variants of our proposed diffusion model with time-mixing and channel-mixing layers for four different tasks. In all cases, except for the conditioning mechanism, the internal architecture remains the same (Supp. Fig. A1). For diffusion timestep and fixed-length conditioning vector, we shift and scale the inputs and outputs to the time mixing and channel mixing blocks using adaptive instance normalization, as done in Peebles and Xie [37].

1. Unconditional generation for synthetic spiking data with Lorenz Dynamics and cortical spiking data in monkeys – we use only time conditioning.

2. Angle-conditioned generation for cortical monkey spiking data – we add an embedding MLP($[\cos\theta, \sin\theta]$) of the reach angle $\theta$ to the timestep embedding output.

3. Trajectory conditioned generation for cortical monkey spiking data – we concatenate the hand velocities $v_x, v_y$ of the monkey with the input as two additional channels.

4. Unconditional variable-length generation for cortical human spiking data – we concatenate the desired length (with a maximum sequence length of 512) as a centered binary mask channel in the input. We only backpropagate through the central section of the output corresponding to the binary mask.

We use a DDPM scheduler with 1000 timesteps and $\epsilon$-parameterization. To stabilize and speed up training, we train all diffusion models using a smooth $L_1$ loss, written as

$$L(x; \delta) = \begin{cases} x^2/(2\delta) & \text{if } |x| < \delta \\ |x| - \delta & \text{otherwise,} \end{cases} \tag{7}$$

with $\delta = 0.05$.

## A2 Dataset access information

All real-world datasets used in this work are publicly available under open-access licenses. Our work does not involve the collection of new experimental data.

The human BCI dataset is available at `https://datadryad.org/stash/downloads/file_stream/2547369` under a CC0 1.0 Universal Public Domain Dedication license. This dataset was originally published in Willett et al. [57]. The data was collected under appropriate ethical oversight, with approval from the Institutional Review Board at Stanford University (protocol #20804).

The monkey reaching dataset (MC_Maze) is available through the DANDI Archive (`https://dandiarchive.org/dandiset/000128`, ID: 000128) under a CC-BY-4.0 license. This dataset contains sorted unit spiking times and behavioral data from primary motor and dorsal premotor cortex during a delayed reaching task.

## A3 Hyperparameters and compute resources

Table 2: Training details for autoencoder models on Lorenz, Monkey reach, and Human BCI datasets. We used the AdamW [31] optimizer, whose learning rate was linearly increased over in the initial period and then decayed to 10% of the max value with a cosine schedule. Mean firing rate for Lorenz was 0.3. In all cases, we used $K = 5$ for the temporal smoothness loss in Eq. 1.

| Parameter | Lorenz | Monkey Reach | Human BCI |
|---|---|---|---|
| **Dataset Details** | | | |
| Num training trials | 3500 | 2008 | 8417 |
| Trial length (bins) | 256 | 140 | 512 (max) |
| Data channels (neurons) | 128 | 182 | 128 |
| **Model Details** | | | |
| Hidden layer channels | 256 | 256 | 256 |
| Latent channels | 8 | 16 | 32 |
| Num AE blocks | 4 | 4 | 6 |
| Spike history | No | Used in unconditional | No |
| **Training Details** | | | |
| Max learning rate | 0.001 | 0.001 | 0.001 |
| AdamW weight decay | 0.01 | 0.01 | 0.01 |
| Num epochs | 200 | 140 (early stop.) | 400 |
| Num Warmup Epochs | 10 | 10 | 20 |
| Batch size | 512 | 512 | 256 |
| $L_2$ reg. $\beta_1$ | 0.01 | 0.001 | 0.001 |
| Temporal smoothness $\beta_2$ | 0.01 | 0.2 | 0.1 |
| CD mask prob. $p$ | 0.2 | 0.5 | 0.2 |

### A3.1 Computational Resources

We performed all training and evaluation of LDNS on the Lorenz and Monkey reach datasets on an NVIDIA RTX 3090 GPU with 24GB RAM. For the Human BCI data, we used an NVIDIA A100 40GB GPU.

The autoencoder for the Lorenz dataset is trained in $\approx 6$ minutes, and the diffusion model in $\approx 20$ minutes. For the evaluation, all sampling is performed on the GPU in 5 minutes. The *effective GPU wallclock time* (time when the GPU is utilized) for the entire training and evaluation run is within 30 minutes.

Table 3: Training details for diffusion models on Lorenz, Monkey reach, and Human BCI datasets. We used the same learning rate scheduler as for the autoencoder.

| Parameter | Lorenz | Monkey Reach | Human BCI |
|---|---|---|---|
| **Model Details** | | | |
| Latent channels | 8 | 16 | 32 |
| Hidden layer channels | 64 | 256 | 384 |
| Num diffusion blocks | 4 | 6 | 8 |
| Num denoising steps | 1000 | 1000 | 1000 |
| **Training Details** | | | |
| Max learning rate | 0.001 | 0.001 | 0.001 |
| AdamW weight decay | 0.01 | 0.01 | 0.01 |
| Num epochs | 1000 | 2000 | 2000 |
| Num warmup epochs | 50 | 50 | 100 |
| Batch size | 512 | 512 | 256 |

For the Monkey reach dataset, the autoencoder with the given hyperparameters is trained in $\approx 8$ minutes, and the unconditional and conditional diffusion models in 40 minutes to 1 hour. With similar sampling times as in Lorenz, the effective GPU wallclock time is approximately within one hour. Optimizing the autoregressive observation model took less than 1 minute.

AutoLFADS, the baseline used for unconditional sampling for the Monkey reach dataset, was trained on a cluster of 8 NVIDIA RTX 2080TI GPUs for one day. As it requires automated hyperparameter tuning to achieve the best accuracy using population-based training (PBT, [24]), AutoLFADS is significantly more compute-expensive to train than LDNS.

For the Human BCI dataset, due to larger trial lengths, more data points, and more heterogeneous temporal dynamics, we trained a slightly larger autoencoder and diffusion model than in Monkey reach. The autoencoder took 50 minutes to train, and the diffusion model took 10 hours to train. Sampling from the trained model took 9 minutes, resulting in a total of under 12 hours of effective GPU wallclock time.

We ran several preliminary experiments for LDNS to optimize the architecture and hyperparameters, as well as for designing appropriate evaluations. We estimate the total effective GPU wallclock time to be $\approx 10\times$ that of the final model runs. The AutoLFADS baseline was only trained once with PBT, as this framework automatically optimizes the model hyperparameters.

We implemented all training and evaluation code using the Pytorch framework[2], and used Weights & Biases[3] to log metrics during training.

## A4   Baseline comparison: Latent Factor Analysis via Dynamical Systems - LFADS

Latent Factor Analysis via Dynamical Systems (LFADS) is a sequential variational autoencoder used to infer latent dynamical systems from neural population spiking activity [51, 36]. LFADS consists of an encoder, a generator, and optionally, a controller, all of which are RNNs. The generator RNN implements the learned latent dynamical system, given an initial condition and time-varying inputs. The internal states of the generator are mapped through affine transformations to lower-dimensional latent factors and single-neuron Poisson firing rates. The encoder RNN maps the neural population activity into an approximate posterior over the generator's initial condition.

At each timestep, the controller RNN receives both encoded neural activity and the latent factors from the previous timestep and outputs an approximate posterior over the input to the generator. The entire model is trained end-to-end to maximize the ELBO, as is done in VAEs. To address the difficulty of hyperparameter optimization for LFADS, Population-Based Training (PBT) has been proposed to automate hyperparameter selection, termed AutoLFADS [25].

---

[2]Paszke et. al. PyTorch: An Imperative Style, High-Performance Deep Learning Library (2019)
[3]Lukas Beiwald. Experiment Tracking with Weights and Biases (2022)

In our experiments with the monkey reach dataset, we use the PyTorch implementation of AutoLFADS [47]. We use the hyperparameters and search ranges from Pei et al. [38], but omit the controller RNN to simplify generation from prior samples. Although this might limit the model's expressiveness, prior research indicates that the monkey reach data can be well-modeled as autonomous, without external inputs from the controller [10]. LFADS has previously performed well on this data without the controller [36].

We generate samples from LFADS by sampling initial conditions from the Gaussian prior, running the generator RNN forward, and Poisson-sampling spikes from the resulting firing rates. For inclusion of spike history in the observation model of LFADS, we used the same training method and hyperparameter settings as in LDNSsh (appendix A1.3).

## A5   Supplementary baseline comparisons

For an extended baseline comparison, we implemented two additional methods for the task of unconditional generation on the Monkey dataset (Sec. 3.4) – Targeted Neural Dynamical Modeling (TNDM, [21]) and Poisson-identifiable VAE (pi-VAE, [63]). It is important to note that while both TNDM and pi-VAE have demonstrated success in analyzing neural and behavioral data, neither was specifically designed for realistic spike train generation. The architectural choices in our implementation of these methods reflect their original intended applications in neural data analysis rather than generation of neural spiking data. Nevertheless, our comparisons show that LDNS, especially with spike-history, is superior or on par with all other methods (Table 4).

### A5.1   Targeted Neural Dynamical Modeling (TNDM)

TNDM [21] is a VAE-based model designed to jointly model neural activity and behavior. TNDM extends LFADS by using an RNN to generate latent dynamics that are mapped to both neural activity and behavioral variables. TNDM separates the latent space into behavior-specific and behavior-independent subspaces to disentangle task-relevant and intrinsic neural dynamics.

For our comparison on the unconditional monkey reach task, we trained TNDM using the architecture and hyperparameters proposed in the original implementation of the paper[4]. We used 64-dimensional latent dynamics for each of the two sets. These project to a total of 10 latent factors $z$ (5 behavior-specific $z_r$ and 5 behavior-independent $z_i$), which is the maximum number demonstrated in the original work.

To generate unconditional samples, we sampled initial generator states from a standard normal prior $\mathcal{N}(0, I)$, then generated the latent dynamics and projected into neuron rates the same way as in LFADS. TNDM, performs well in matching real data in spike statistics (Supp. Fig. A13 cyan, Supp. Fig. A16e) and temporal dynamics (Supp. Fig. A15). Overall, we observe that LDNS captures spike statistics better than TNDM, except for the the population spike history count. In all metrics, LDNS augmented with spike history outperforms TNDM on spike statistics.

### A5.2   Poisson-identifiable VAE (pi-VAE)

Poisson-identifiable VAE (pi-VAE) [63] is a VAE-based model for count data that ensures identifiability in the latent space. pi-VAE does **not** model temporal dependencies, instead treating each time point as an independent sample.

We trained pi-VAE on the monkey dataset using the original architecture and hyperparameters[5]. We used a General Incompressible-flow Network as a decoder, with 2 behaviorally relevant dimensions and 2 independent dimensions in the latent space. However, our evaluation context differs significantly from the original paper's demonstrations: while pi-VAE was initially evaluated on 50ms time bins and straight reaches only, our comparison uses 5ms bins and conditions on angles across all reaches at both middle and end trajectory points. The "label", or behavior, is presented as a 4-dimensional vector containing the cosine and sine of initial and final reach angles. Since this has a conditional latent space, sampling is performed by sampling angles randomly.

---

[4]Code adapted from Github respository: https://github.com/HennigLab/tndm.

[5]Code adapted from article by Lyndon Duong (2021) – https://www.lyndonduong.com/pivae.

Importantly, sampling from pi-VAE does not introduce any temporal dependence between spike bins within a trial — pi-VAE was not intended to be a generative model of neural spiking data. The lack of temporal modeling in pi-VAE's is a fundamental limitation for generating realistic spike trains, as evident in our empirical comparisons (Supp. Fig. A13 in yellow, Supp. Fig. A15,A16f). Note that this failure cannot be diagnosed simply from looking at the sampled spiking data (Supp. Fig. A14).

### A5.3 Contributions of LDNS in context to LFADS, TNDM and pi-VAE

- LDNS is designed specifically for the purpose of accurately generating neural spiking data (unconditionally or conditionally)—a task often ignored by other LVMs designed for neural data analysis such as LFADS, pi-VAE, and TNDM.

- The S4 autoencoder and diffusion model in LDNS are trained in separate stages, offering modularity, while both components naturally account for temporal dependencies (unlike pi-VAE).

- S4 is autoregressive, similar to other RNN-based models, but empirically we found it to perform better when extending past the training trial length (compared to LFADS, see Sec. A6.1).

- One feature provided by some neural-behavioral analysis models (such as pi-VAE and TNDM) is an explicit disentangling of neural vs. behavior-relevant latents. While we found LDNS latents contain relevant behavioural information (Fig. 5e-g, Supp. Fig. A17), we did not explicitly supervise the latent space to induce this property.

- Finally, the spike history-dependent observation model in LDNS is modular and can be optimized post-hoc using rate predictions of any model to improve spike generation quality. We observed this with LDNS as well as LFADS (Table 1).

Table 4: **Added Baselines metrics comparison.** $D_{KL}$ for the population spike count histogram and RMSE comparisons. Mean and standard deviation across 5 folds sampled with replacement. **Bolded** entries represent best-performing values for Poisson observation and spike-history observation model, respectively.

| Method | $D_{KL}$ psch | RMSE pairwise corr | RMSE mean isi | RMSE std isi |
|---|---|---|---|---|
| pi-VAE | $0.0063 \pm 2.9$e-4 | $0.0031 \pm 1.08$e-5 | $0.064 \pm 0.002$ | $0.034 \pm 0.001$ |
| TNDM | $\mathbf{0.0028 \pm 6.6}$**e-5** | $0.0027 \pm 1.17$e-5 | $0.057 \pm 0.004$ | $0.029 \pm 0.001$ |
| AutoLFADS | $0.0040 \pm 2.2$e-4 | $0.0026 \pm 1.25$e-5 | $0.039 \pm 0.003$ | $0.029 \pm 0.001$ |
| LDNS | $0.0039 \pm 3.9$e-4 | $\mathbf{0.0025 \pm 1.1}$**e-5** | $\mathbf{0.037 \pm 0.001}$ | $\mathbf{0.023 \pm 0.001}$ |
| AutoLFADSsh | $0.0036 \pm 2.1$e-4 | $0.0026 \pm 1.8$e-5 | $0.034 \pm 0.002$ | $\mathbf{0.023 \pm 0.0001}$ |
| LDNSsh | $\mathbf{0.0016 \pm 6.2}$**e-4** | $\mathbf{0.0025 \pm 1.07}$**e-5** | $\mathbf{0.024 \pm 0.002}$ | $\mathbf{0.023 \pm 0.001}$ |

## A6 Supplementary Figures Lorenz

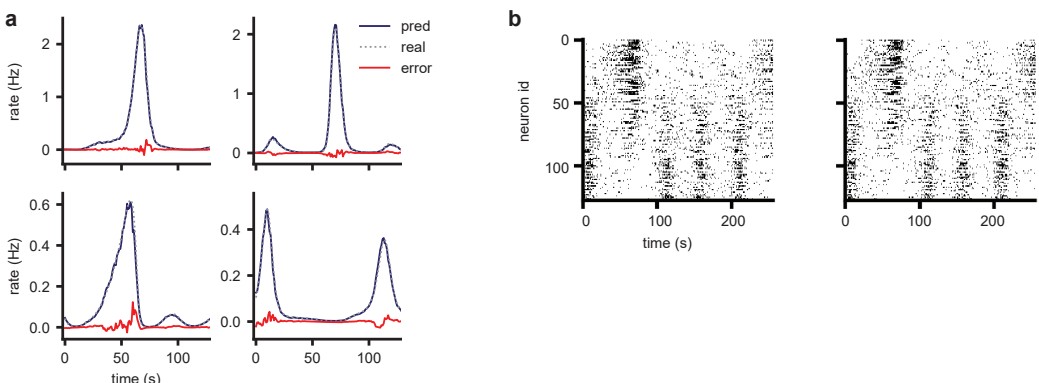

Figure A2: **The autoencoder captures the gt Lorenz synthetic firing rate perfectly a)** Autoencoder predictions (pred) and true rates from the test set, together with their difference (error, in red). **b)** Reconstructions sampled from the Poisson observation model (right) closely resemble the test sample (left). Both spiking activity is binarized for the visualization.

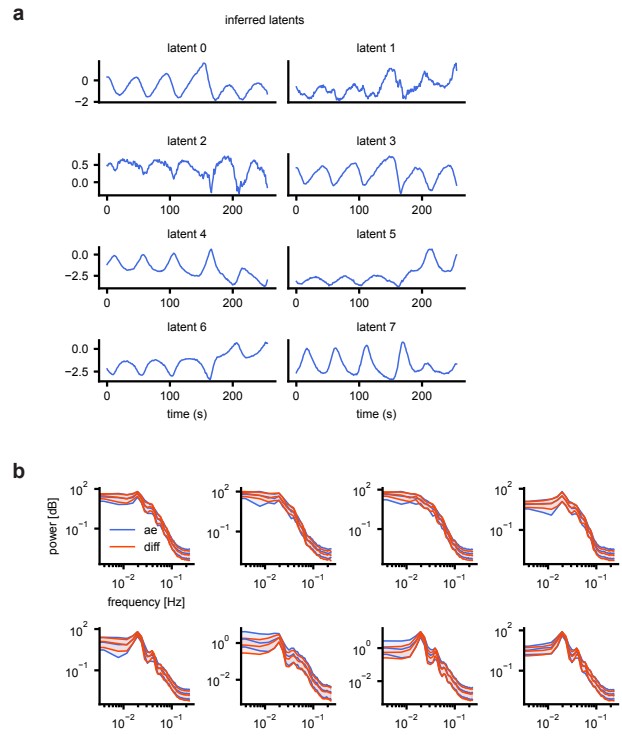

Figure A3: **The S4 autoencoder infers smooth latents from discrete spikes and samples from the diffusion model capture the latent distribution. a)** Inferred autoencoder latents for a test sample. **b)** Power spectral density for all eight latent dimensions for the inferred autoencoder training set and samples from the diffusion model

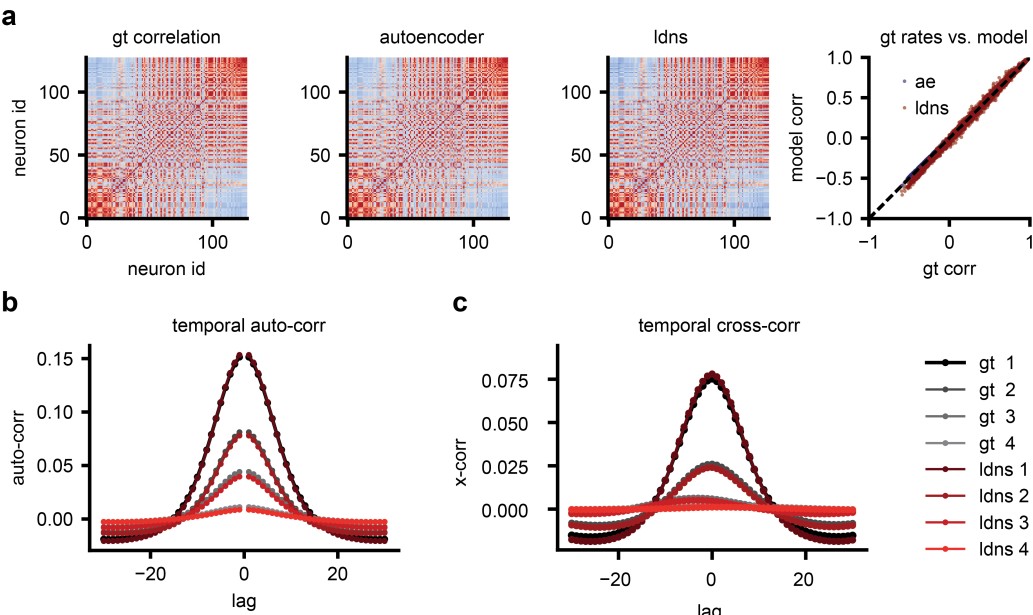

Figure A4: **LDNS captures the correlation structure of the Lorenz dataset a)** Both the autoencoder and LDNS-sampled rates capture the ground truth instantaneous correlation structure of the synthetic rates. **b)** The auto-correlation structure of ground truth and sampled spiking activity matches perfectly in 4 neuron groups, sorted according to correlation strength. Synthetic Lorenz data group $x$ is denoted by gt $x$, LDNS samples as ldns $x$. **c)** The time-lagged cross-correlational structure is also perfectly captured by LDNS in all groups.

### A6.1 Length generalization of LFADS on Lorenz

To analyze whether LFADS [36] exhibits similar length generalization properties as LDNS, we trained an AutoLFADS model on the Lorenz dataset (256 bins). We used the same architecture as the Monkey dataset, with 40-dimensional latent dynamics. We sampled initial conditions from the LFADS prior, then generated dynamics for both the original 256 steps and for an extended duration of $16\times$ the training length.

Qualitatively, we observed that while LFADS produced trajectories resembling the attractor dynamics of the ground truth Lorenz system (Supp. Fig. A5a, across various dimension combinations), these trajectories often diverged when run for longer intervals (Supp. Fig. A5b). However, the system eventually returned to typical dynamics.

Furthermore, when generating for extended durations, we observed that the mean population firing rates sometimes reached extreme values in some samples (Supp. Fig. A6), though they eventually returned to typical ranges. This behavior was not observed in LDNS samples, suggesting that bidirectional generation in the diffusion model provides more stability in variable-length generation.

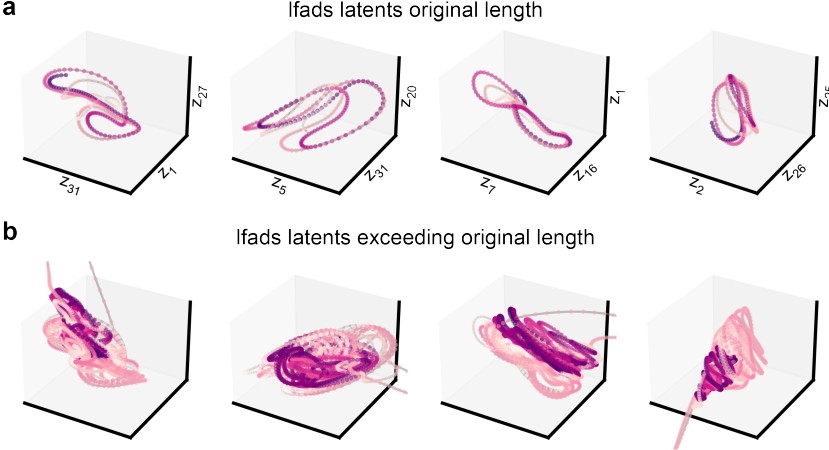

Figure A5: **Length generalization of LFADS on Lorenz** Different projections of a 40-dimensional latent space from LFADS trained on the Lorenz system. Trajectories are compared between **a)** the original length and **b)** 16 times the original length using sampled initial conditions. For comparison with LDNS length generalization, see Fig. 2c

.

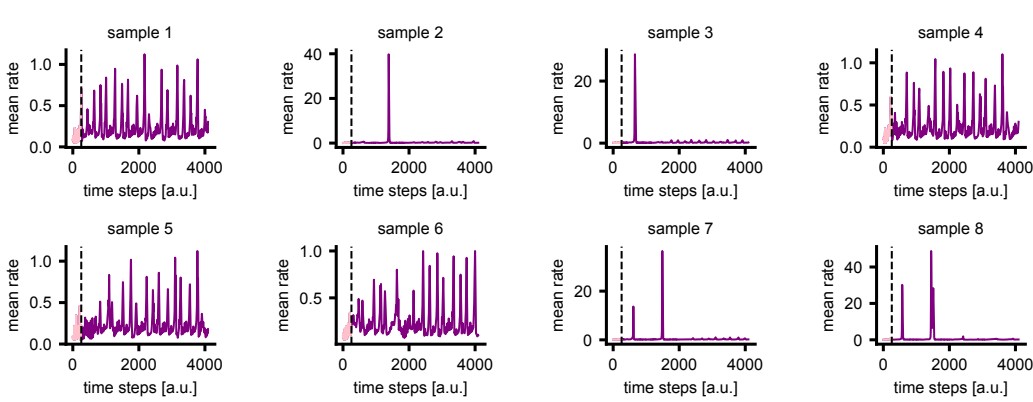

Figure A6: Mean population firing rates for eight different samples, shown for both the original length (pink) and $16\times$ the original length (purple).

# A7 Supplementary Figures Human BCI

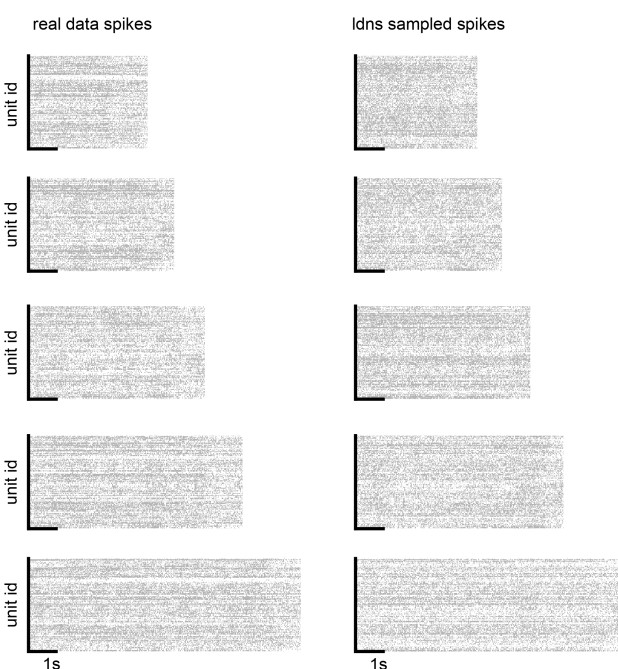

Figure A7: Visual comparison of different sampled spiking data from LDNS, with five samples from the true dataset.

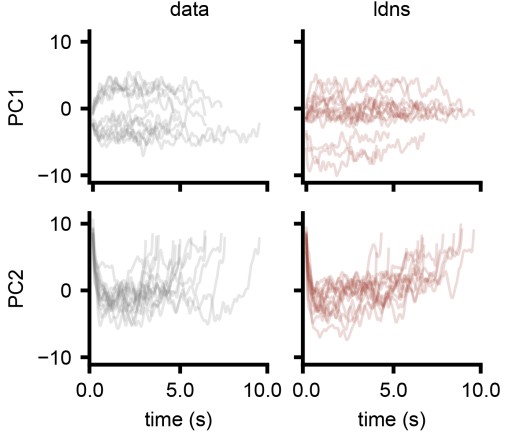

Figure A8: First two principal components (PCs) of smoothed spikes from true data (left) and model samples (right). Each line represents one sampled trial. Spikes were smoothed using a Gaussian filter with a window of 160ms prior to extracting the PCs.

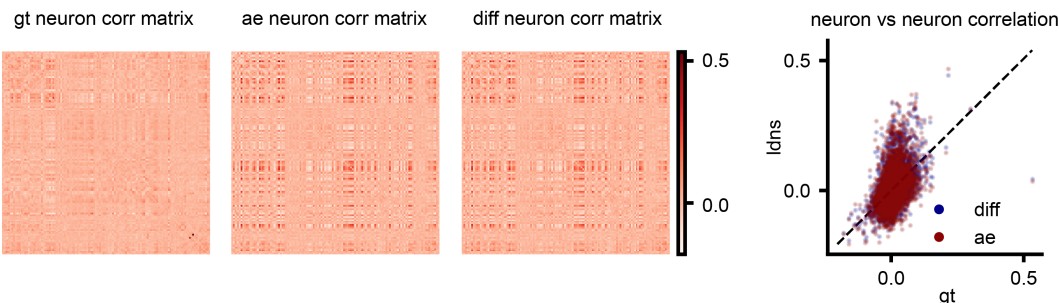

Figure A9: Correlation matrices for real spiking data from human and LDNS, comparing the autoencoder-inferred (ae) correlation (sampled from reconstructed rates) and correlation of sampled spikes (diff). The deviations from the data (gt) already arise at the autoencoder stage.

# A8  Supplementary Figures Monkey Reach task

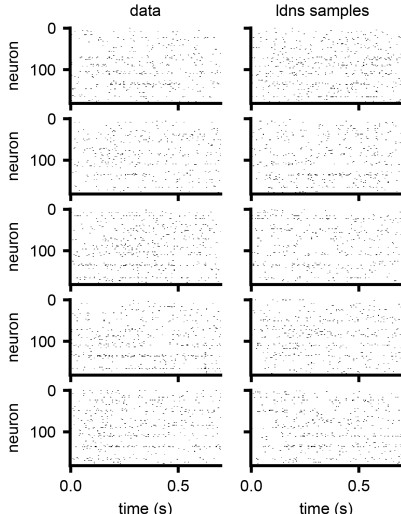

Figure A10: Correlation matrices for real spiking data and samples from Poisson LDNS, concatenated across trials, comparing the autoencoder-inferred (ae) correlation (sampled from reconstructed rates) and correlation of sampled spikes (diffusion + ae). Most deviations from the data (gt) already arise at the autoencoder stage.

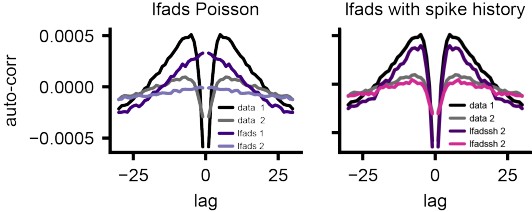

Figure A11: Visual comparison of different sampled spiking data from LDNS with five samples from the real dataset.

Figure A12: **Equipping LFADS with spike history** Auto-correlation of data, LFADS samples with Poisson observations (left) and LFADSsh samples (with spike history), grouped according to correlation strength.

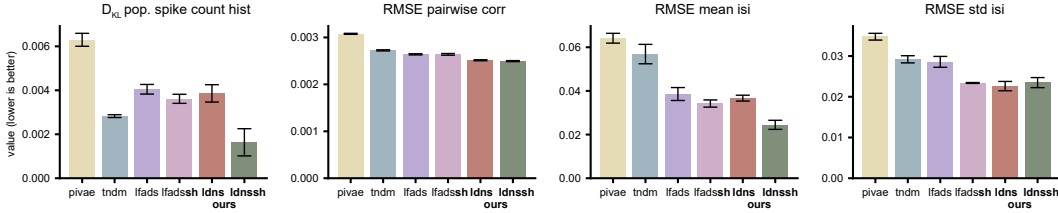

Figure A13: **Performance comparison with additional baselines** pi-VAE [63], TNDM [21], LFADS [36], LFADS with spike history (LFADSsh), LDNS and LDNS with spike history (LDNSsh). Mean and standard deviation across 5 folds sampled with replacement.

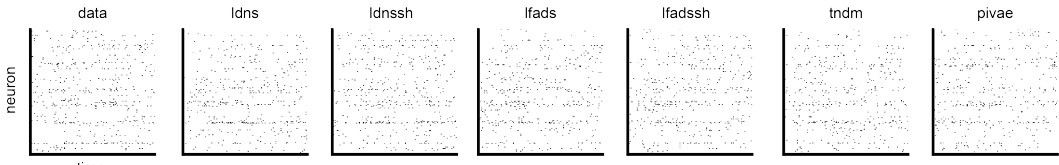

Figure A14: Visual comparison of sampled spiking data from LDNS and all baselines with real data.

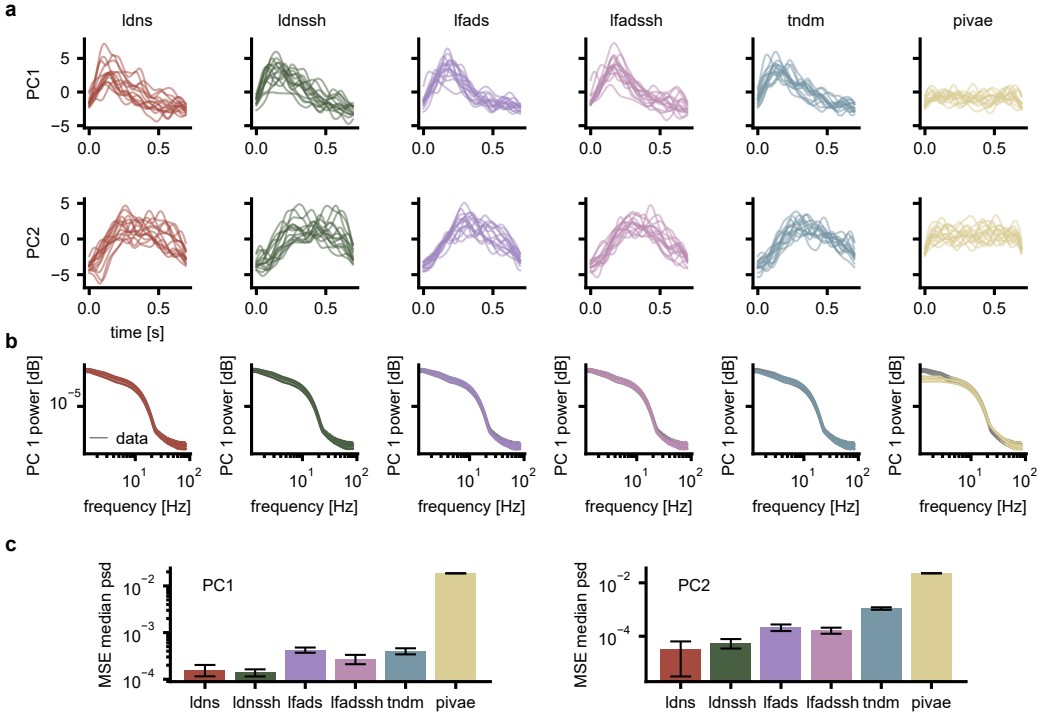

Figure A15: **Comparing principal components of smoothed sampled spikes a)** First two principal components (PCs) of smoothed spikes from model samples. Each line represents one sampled trial. Spikes were smoothed using a Gaussian filter with a window of 40ms prior to extracting the PCs. The PCs were fit using real data. Since pi-VAE does not account for temporal dynamics, it does not show any temporal structure in the PCs. **b)** Power spectral density (PSD) of PC1, plotted for model vs. data (in grey). **c)** Mean squared error of median PSD between model samples and data for PC1 (left) and PC2 (right). LDNS and LDNSsh perform the best here, with pi-VAE showing large errors.

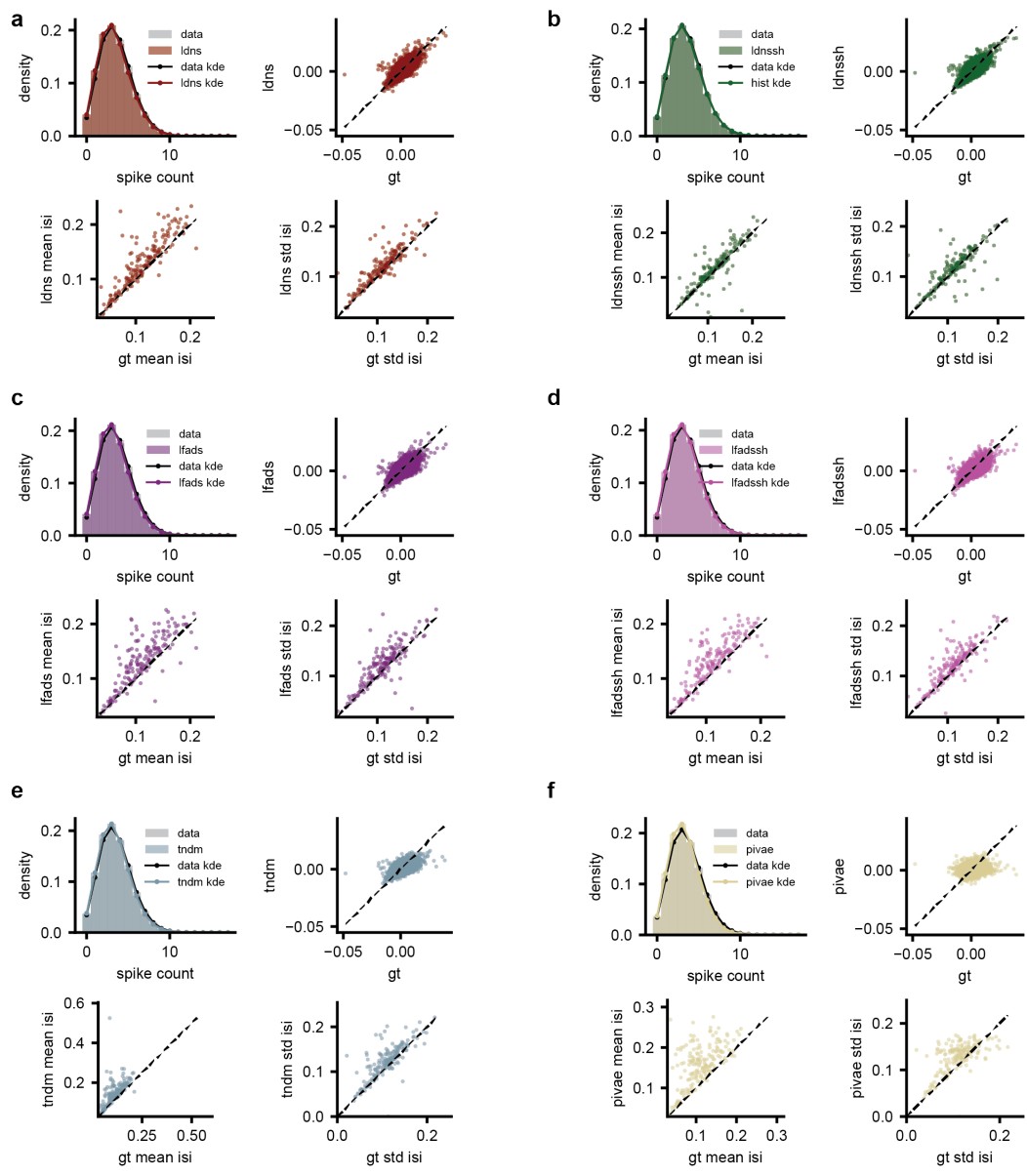

Figure A16: Population-level and single neuron-level statistics of **a)** LDNS, **b)** LDNSsh (with spike history), **c)** LFADS, **d)** LFADSsh (with spike history), **e)** TNDM, and **f)** pi-VAE.

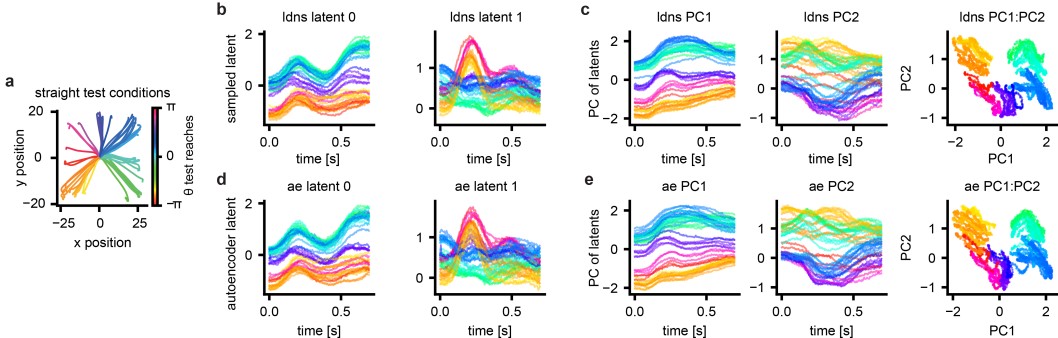

Figure A17: **Comparison of LDNS latent space trajectories of inferred and conditionally sampled latents a)** Straight reaches from the Monkey reach test set. **b)** LDNS sampled latents (velocity-conditioned on reaches shown in a). **c)** PCs of LDNS sampled latents. **d)** Autoencoder-inferred latents of corresponding neural activity for the reaches shown in a). **e)** PCs of autoencoder-inferred latents.

