# OpenReview forum: "Latent Diffusion for Neural Spiking Data"
_NeurIPS.cc/2024/Conference — NeurIPS 2024 spotlight_

### Official Review · Reviewer_mYEj · 2024-07-05

**Soundness:** 3
**Presentation:** 4
**Contribution:** 2
**Rating:** 7
**Confidence:** 4

**Summary:**

In their study "Latent Diffusion for Neural Spiking Data", the authors introduce the titular LDNS model for generating realistic neural population activity, and apply it to three datasets: a synthetic dataset with true latents generated from the three-dimensional Lorenz system, and two previously published neuroscientific datasets.

The LDNS model is an instance of the latent diffusion model (LDM) introduced by Rombach et al. (2022), which is adapted here by the authors to support generation of variable-length discrete-valued time series. The autoencoder of the LDM in this case is trained with a Poisson loss to account for the discrete nature of neural spiking data. The diffusion model of the LDM operating in latent space here is constructed with layers of structured state space sequence (S4) models of Gu et al. (2021). The diffusion model can be trained to generate samples conditional on certain external covariates by conditioning the reverse mode on the covariates. The split of LDMs between the autoencoder stage and the diffusion stage is particularly convenient for the task of generating spike trains, as it avoids any adjustment to the diffusion model for generating discrete variables. A variant of the LDNS model has additional parameters trained post-hoc to incorporate the spike-time history of individual output neurons to account for refractoriness and other single-cell auto-regressive features.

According to the considered neural population statistics, the LDNS model gives an excellent fit to the synthetic data generated from the three-dimensional Lorenz system. For the two neuroscientific datasets, the LDNS performs solid in terms of matching the considered neural population statistics, outperforming the important reference model LFADS (Sussillo, 2016) in one case. On the first neuroscientific dataset comprised of human neural population data sampled during attempted speech, the LDNS is shown to handle variable sequence lengths in the training data and during sampling. The second neuroscientific dataset consists of monkey neural population data recorded during a maze reach task. Here, the LDNS variant with additional spike-history parameters fitted post-hoc is shown to also capture the temporal auto-correlation structure of neural subpopulations. The authors furthermore showcase the ability to generate samples conditionally on experimental covariates, by conditioning sampling on the initial reach angle, respectively the full velocity trajectory.

**Strengths:**

The submitted work is original, in that latent diffusion models (LDMs) to my knowledge have not yet been used to (conditionally) generate neural population activity. I agree that LDMs are indeed a nice idea here, since the separation into autoencoder stage and diffusion model stage means we can try to use denoising diffusion models without much changes.

The work seems technically solid to me. The authors make sensible changes to the LDM for the task of generating neural population activity. The adaptations for discreteness (Poisson loss, spike-history terms) show a good understanding of best practices in neural data analysis. The inclusion of S4 layers into autoencoder and diffusion models to handle and generate variable-length sequences is interesting, and could further spread usage of these tools in the computational neuroscience community.

The manuscript is well written, explaining not only the key ideas and experiments, but also summarizing the important model parts (autoencoders, denoising diffusion models) next to the non-standard adaptations (Poisson loss, S4 layers, spike-history terms) done by the authors for this study. The figures are high quality and way above what I would expect from a conference paper.

This study follows and expands a (small) recent trend of using modern machine learning algorithms for generation of neural population activity. LDNS models / LDMs clearly have some advantages over previously suggested models, in that they combine the low-dimensional representation of autoencoders with the generative fidelity of diffusion models.

**Weaknesses:**

The authors have adapted the latent diffusion model to the task of generating neural population activity. While I believe they did good work on that, I am less convinced of the evaluation and comparison of their model, which is important to judge its overall usefulness.

For a generative model of multivariate time-series, I find the evaluation somewhat lacking. I understand that judging spike train generation is still a much harder task than judging image generation, and that one could argue that to date we don't understand all the relevant aspects of neural population activity. But firing rates, pairwise correlations and population spike counts are features that don't take dynamics into account. Average interspike interval and standard deviations of interspike intervals are still very local temporal features. Computational neuroscience has come up with a host of methods for analysing neural population dynamics -- in particular those with low-dimensional structure as assumed by the LDNS model -- which could be used here for comparing the sampled spike trains against the data. The authors already cited both LFADS and Gaussian Process Factor Analysis, but did not compare e.g. the low-dimensional trajectories extracted from their sampled data against the low-dim. trajectories extracted from the neural recordings. If nothing else, a supplementary figure or two with a handful of additional sampled spike trains against real recordings could help readers judge if the model captures the overall neural dynamics of the data.

Another aspect that makes it difficult to evaluate the quality of their suggested model is the lack of comparison against direct competitors. The authors do an excellent job of explaining LFADS and its relevance as a comparison, but from what I understand only compare against LFADS on one of the three numerical experiments (I take that figure A4 which isn't mentioned in the text is also about the monkey data?). I understand that autoLFADS is significantly more computationally expensive to run than the LDNS model,  (which I would encourage the authors to state more prominently also in the main text!) but especially for the second experiment with human data, the LDNS model seems to perform the worst of the three experiments, and I could find no alternative model for comparison.

**Questions:**

As explained above, analysis of the full population dynamics of their sampled neural population data would be very interesting. The authors could, but do not have to follow my suggestion of analysing the sampled neural population with one of the more commonly used dynamical dimensionality reduction methods.

Can we get some comments or additional analysis of the LDNS model in cases where it doesn't work perfectly? There are several interesting supplementary figures for the Lorenz synthetic data, but for the other two experiments I am left wondering whether the non-negligible model errors arise primarily from the autoencoder or from the diffusion model. Those are the cases where I would most appreciate a comparison of the pairwise correlations from autoencoder against those of the full LDNS model, as in Fig A7 for the Lorenz data.

**Limitations:**

I think the authors have adequately addressed limitations and possible negative societal impacts of their work.

---

> ### Author Rebuttal · Authors · 2024-08-06
>
> We thank the reviewer for their detailed summary and evaluation and suggestions that will significantly improve our work, e.g. recommending to further assess the dynamics of the generated samples. We are also thankful for the generally positive response and describing our work as original.
>
> **Analyzing population dynamics to assess sample quality beyond spike statistics**
> We agree with the reviewer’s point that an evaluation of LDNS-generated samples in terms of dynamics would be valuable. We addressed this in two different ways:
>
> **1\. Principal Component Analysis (PCA) on smoothed spikes:**
> We followed a common approach in neuroscience of performing dimensionality reduction using **PCA on smoothed spikes and then visualizing the resulting first n principal components** over time (**Fig. R7; please see attached PDF**, here, n=2). Overall, we see that PCs of unconditional LDNS samples closely match that of the real held-out data samples. This approach allowed us to clearly see inconsistencies in the dynamics, e.g. for an added baseline model pi-VAE (see Reviewer AuW4) that does not model dynamics, leading to flat PCs (**Fig. R7, left, yellow**) and higher MSE between the median power spectral density of pi-VAE samples to true data as compared to all other methods (**Fig. R7, right**).
>
> **2\.  Comparing inferred latent dynamics of sampled spikes using LFADS:**
> As a more sophisticated dynamics analysis, we followed the reviewer’s suggestion and embedded both the true data and the unconditionally generated sampled spiking data from LDNS using **LFADS** (used here as an **analysis tool**). This allows us to compare latent dynamics by comparing the distributions of inferred initial states of the LFADS generator RNN (**Fig. R8**, visualized as PCs of $g\_0$) across true data and generated spikes. This analysis revealed that LDNS spikes more closely capture the broad true data distributions than, e.g., spikes sampled from LFADS (as a generative model) (**Fig. R8**). Such analyses, however, require a well-fit state-space model, which in of itself poses a challenge for many datasets.
>
> We agree on the **value of visualizing more spike raster plots and extracted principal components**, and will add them in the Appendix for all models and tasks. Similar to Appendix A6, we will also add and quantify power spectral densities of the principal components (**Fig. R7**) for the different methods against the real data.
>
> **Fitting the LFADS baseline to the human data and the Lorenz dataset**
> Following the reviewer’s suggestion, we now also trained LFADS on the human dataset and the Lorenz dataset.
>
> **Human:** The human dataset is challenging as the trials are highly heterogeneous with different lengths. LDNS can handle these challenges due to our architectural choices and our masking training scheme even if, as pointed out, LDNS does not work as perfectly (see discussion below) as in the other datasets. Please note that this is a new dataset (released less than a year ago), and to date, no readily available generative modeling baseline exists for this data. To allow for a fair comparison, we now attempted to modify LFADS so that it can be fit to this data.
>
> To be able to fit LFADS on this dataset, we had to cut the data into equal length segments of 140 time steps (2.8s), since without major modifications LFADS (with its bidirectional encoder architecture) is not well-suited to handling variable length inputs. Despite capturing the spiking statistics on the 2.8s length it was trained on \-- LFADS failed to capture any realistic dynamics beyond the 2.8s cut-off during sampling (**Fig. R3**). The latents of LFADS decay to a fixed point when the RNN is run forward beyond 2.8s, indicating that it is ill-suited to variable-length generation of such heterogeneous data (**Fig. R3**), highlighting the need for flexible methods such as LDNS.
> We acknowledge that there may be variants of LFADS that can handle this case better, however, our point is that “vanilla” LFADS is unable to handle this task without significant modifications.
>
> **Lorenz:** While it is well established that LFADS fits the Lorenz dataset well (Sussillo et al. 2015, Pandarinath et al. 2018), LFADS struggles with length generalization: Running the LFADS generator on 16 times the original length results in inconsistent latent trajectories (**Fig. R2**) compared to LDNS (Fig. 2c).
>
> We will additionally point out in the main text, not just in the Computational Resources Appendix, that LFADS is significantly more computationally expensive to run than LDNS, and we thank the reviewer for this suggestion.
>
> **Comparison against additional baselines on the monkey reach task**
> In addition, as also suggested by other reviewers, we include two additional baselines for comparison on the monkey dataset: TNDM (Hurwitz et. al, 2021\) and pi-VAE (Zhou et al. 2020), are VAE-based models that were proposed as analysis tools to jointly study neural and behavioral data. We find that LDNS and its spike-history based extension outperforms these baselines on the monkey dataset (see **Fig. R1** and response to reviewers AuW4 and pMhd for implementation details).
>
> **Contribution of autoencoder vs. diffusion for correlations**
> We agree that disentangling the contribution of the S4 autoencoder and diffusion is useful, particularly in cases when LDNS does not work perfectly. The model errors arise primarily from the autoencoder, not from the diffusion model: The values of pairwise correlations both in monkey reach data and human are very similar for the autoencoder reconstructions and LDNS samples (**Fig. R9**)**,** i.e. mismatches arose at the autoencoder and not the diffusion stage.
>
> Analogous to the Lorenz experiment (Fig. A7), we will add the analyses comparing the autoencoder performance and the diffusion performance in the Supplementary Material for the two other datasets (monkey, human).

---

> > ### Comment · Reviewer_mYEj · 2024-08-13
> >
> > I thank the authors for their in-depth response and the additional analyses they added to address my concerns.
> > I am happy to raise my rating by a point.

---

> > > ### Author Response · Authors · 2024-08-14
> > >
> > > Thank you for the positive response, detailed engagement with our work, and further increasing the score.

---

### Official Review · Reviewer_pMhd · 2024-07-06

**Soundness:** 2
**Presentation:** 3
**Contribution:** 3
**Rating:** 6
**Confidence:** 3

**Summary:**

This paper introduces the Latent Diffusion Model for Neural Spiking data (LDNS). LDNS combines the capacity of autoencoders to extract low-dimensional representations of discrete neural population activity with the capability of denoising diffusion probabilistic models to generate realistic neural spiking data. It achieves this by modeling the inferred low-dimensional continuous representations. Through experiments on three different datasets, LDNS has been proven to achieve low-dimensional latent variable inference and realistic conditional generation of neural spiking datasets, providing possibilities for simulating experimentally testable hypotheses.

**Strengths:**

1. Quality: The paper is comprehensive, with detailed explanations of the model's characteristics and corresponding experimental designs.
2. Clarity: The paper is logically clear and well-structured.
3. Importance: The paper significantly contributes to addressing the modeling challenges of complex datasets in neuroscience.

**Weaknesses:**

1. There is only one comparison method in this paper.

**Questions:**

1. What is the significance of studying the generation of spiking data? Is this generated data of practical value?
2. Have you tried to compare it with other VAE-based methods? Despite the claim of the authors that LFADS is the most successful VAE-based method available, it is clear that no method performs best in all tasks, especially in more cutting-edge tasks like this paper.
3. Can you explain how the stochastic operations in Figure 1 are performed?
4. In the conditional generation experiment of neural activity given the reach direction, the experiment is conducted under conditions of only two directions. Is it possible to conduct the experiment under conditions of three directions?

**Limitations:**

This paper still has certain limitations. Firstly, the exploration of LDNS in simulating neural activity is currently confined to the abdominal cortex during speech, which leaves the simulation of neural activities under more complex behavioral patterns in other parts of the body unexplored. Secondly, the processing of human-related data in the research raises concerns about privacy protection. In response to these limitations, the following suggestions are offered:
1. This paper has initiated an investigation into the effects of LDNS in simulating neural activities in the abdominal cortex during human speech. To further verify its potential, future research could expand to simulate cortical neural activities in other parts of the body under more complex behavioral patterns, aiming for a comprehensive evaluation of LDNS's applicability and efficacy in the field of neuroscience.
2. Regarding the use of human-related data in the paper, it is suggested that in subsequent research, greater emphasis should be placed on protecting data privacy to avoid negative social impacts.

---

> ### Author Rebuttal · Authors · 2024-08-06
>
> We thank the reviewer for finding our work comprehensive and our writing and presentation clear and well-structured. We also appreciate that the reviewer finds our contributions of addressing the modeling challenges of complex neuroscientific datasets significant.
>
>
> **Additional comparison with other VAE-based methods**
> Following reviewer suggestions (as well as that of AuW4), we have now included two additional proposed VAE-based models as baselines:  Poisson-identifiable VAE (pi-VAE, Zhou et. al, 2020\) and Targeted Neural Dynamical Modeling (TNDM, Hurwitz et. al, 2021\) on the unconditional monkey reach task. We emphasize that while these LVMs have successfully been used for analyzing neural and behavioral data, they were not intended for realistic spike generation.
>
> New figures (**Fig. Rx**) are in the **PDF**.
>
> Compared to these additional baselines, as well as an extended LFADS model augmented with spike history, we found that LDNS remains superior in realistic spike generation (**Fig. R1**), and recovers structured latents informative of behavior (**Fig. R6**)**.** Implementation details of these baselines are provided below, and will be included in the revised paper.
>
> We also now trained LFADS on the Lorenz and human BCI tasks, in particular to evaluate its ability to length-generalize and handle variable length input data compared to LDNS. We find that LFADS had difficulties in extending the learned dynamics correctly (**e.g., Fig. R2,3**) – in contrast to LDNS, which can accurately generalize to 16 times the training length in the Lorenz task and can naturally deal with variable length trials in the human task.
>
> **TNDM**: We trained TNDM on the monkey dataset using the original proposed architecture and model hyperparameters. We used 5+5 latent factors, the maximum shown in the original paper. We used the prior N(0,1) to sample the initial generator states for unconditional sampling of spikes (**Fig. R1,** blue).
>
> **pi-VAE**: We trained pi-VAE on the monkey dataset using the original proposed architecture and model hyperparameters. pi-VAE’s architecture does not consider temporal dynamics, and treats each time point as an independent sample. Furthermore, while Zhou et al. 2020 evaluated pi-VAE on 50ms time bins and straight reaches only, we here use 5ms bins and condition on angles of all reaches in the middle and end of the trajectory. Sampled spiking data shows poor statistics mainly due to the lack of temporal dependence in the model (**Fig. R1, R7,** yellow).
>
> **Significance and practical value of realistic spiking data generation**
> Accurate modeling and generation of neural spiking data has scientific, clinical, and practical value. If generated data were used for augmenting training for a downstream task, for example, introducing obvious artifacts (such as not capturing refractory periods) can introduce bias. Furthermore, when studying relations between variables in neuroscience with such emulator models, subtle changes such as spike times or phases when spikes occur in oscillations are known to make a difference.
>
> **Modeling activities from other brain regions**
> Our work proposes a general methodology for modeling and generating spiking data, and is agnostic to the particular brain region where it may be recorded from. In our evaluation, we have considered the motor cortex of monkeys and the speech cortex of humans, but it can be straightforwardly applied to other datasets as well. We will discuss these possibilities in the revised paper.
>
> **Clarification on ethics concern and suggestions on data privacy**
> The human BCI dataset we used is publicly available under a CC0 1.0 Universal Public Domain Dedication license. It is from a peer-reviewed paper that was previously published in Nature (Willett et al. (2023). A high-performance speech neuroprosthesis. *Nature*.), and according to this paper, was cleared in ethical reviews by the Institutional Review Board at Stanford University (protocol \#20804).  Our paper did not provide new sensitive data, nor does it provide a methodology to obtain such data. We will make this information clear in the appendix of the revised paper.
>
> We agree that protecting data privacy is very important, especially when using sensitive data involving human participants, and will further acknowledge this in the Discussion section.
>
> **Other clarifications**
> **Stochastic operation in Figure 1**: Unless specified otherwise, we use the Poisson observation model as the stochastic operation, going from inferred Poisson rates to spike counts. In specific scenarios (e.g., Fig. 4e), we extend this observation model to include spike history dependence in the LDNSsh model. This dependence allows us to capture e.g. refractory periods of neuron firing behavior by reducing the probability of a spike occurring directly after a previous spike, enabling LDNSsh to accurately capture biologically plausible spiking statistics.
>
> **3-axis reach conditioning**: Our architecture is agnostic to the number of reach axes. The dataset we consider involves monkeys performing 2-dimensional reach tasks, which we pass to the diffusion model as two additional channels or as a scalar reach angle (see Fig. 5 and A1). This can be extended straightforwardly to condition on higher-dimensional behavioral variables, allowing to also model reaches in 3-dimensions.
>
> We hope these additions, in particular the inclusion of additional comparison methods, and clarifications, address the reviewer’s concerns and enable them to raise their score.

---

### Official Review · Reviewer_y7Q4 · 2024-07-08

**Soundness:** 2
**Presentation:** 3
**Contribution:** 3
**Rating:** 7
**Confidence:** 4

**Summary:**

This paper proposes a new generative model for neural spiking datasets. The model consists of a deterministic, deep SSM (S4) autoencoder paired with a diffusion model of the learned autoencoder latent sequences. This enables generating accurate neural time series traces across variable length trials lasting up to 10ms and conditional generation given behavioral covariates. The approach is applied to a synthetic dataset and two different neural datasets, where the authors investigate a variety of uses of the model.

**Strengths:**

The proposed generative model of neural activity appears powerful and generally useful of fitting neural responses across a large variety of conditions. Both the unconditional and conditional generative performance of the model is impressive. Many components of the model and training process are well-motivated and clearly described. For these reasons, I find this to be a significant contribution.

**Weaknesses:**

While much of the modeling approach is well-motivated, I do not find that to be the case for the specific implementation of the spike-history component. Additionally, it appears that the spike-history component is responsible for much of the improvement over LFADS, and not the alternative underlying generative model that is the primary novel contribution of this paper (one could imagine also training LFADS with a spike history filter). In particular, it is not well-motivated why the filters are trained post-hoc and why the softplus approximation is preferred over using the exponential function.

**Questions:**

- Why is the spike-history filter trained posthoc instead of during the autoencoder training? Additionally, can the authors provide more details about why they choice the softplus approximation? I do not necessarily agree that the approximation is accurate, and its not clear why this approximation is even necessary during training or why it is used over a numerically stable exponential function. Is this primarily an issue when the model is run in generative mode?

- The LDNS overestimates some of the pairwise neural correlations in the attempted speech dataset. Is it typical for LDNS to overestimate pairwise correlations? Could this be improved by increasing the latent dimensionality of the model, to allow for more uncorrelated latent dimensions?

- Could the authors comment on the use of S4 as compared to other deep state space models like Mamba? Does one appear to work better than the other?

- Have the authors considered using the approach to do conditional generation of the attempted speech neural recordings given the cued sentence? This seems like a challenging but direct application of the proposed method.

**Limitations:**

Yes.

---

> ### Author Rebuttal · Authors · 2024-08-06
>
> We thank the reviewer for finding the contributions presented in our work significant, and for noting the flexibility and performance of our model in generating unconditional and conditional neural spiking data in a wide variety of conditions.
>
> The reviewer had questions and concerns over training and architectural details of the model, and the role of the spike history dependence model in the superior performance of LDNSsh. To disentangle the contribution of this observation model from the architectural contributions of our work, as suggested by the reviewer, we now trained an extended LFADS model by equipping it with the same spike history observation and found that it is still matched or outperformed by the equivalent LDNSsh model. We also discuss our reasoning for fitting these terms post-hoc, and why we used a softplus approximation. Finally, we address the reviewer’s questions on architectural choices, and on the attempted speech dataset.
>
> New figures (**Fig. Rx**) are in the **PDF**.
>
> **Equipping LFADS with spike history**
> We agree that a better characterization of the contribution of spike history, relative to S4 and diffusion, would be beneficial. We have now extended LFADS with spike history dependence (LFADSsh) using the approach we introduced for LDNS, and found it to improve its performance (**Fig. R1,4**). However, LDNS with spike history (LDNSsh) is still superior or on par on spike generation metrics.
>
> Furthermore, even without spike history, LDNS was already on par or better compared to its counterpart in LFADS in our original evaluation, and we will emphasize the corresponding comparisons more prominently in Table 1 (LDNS vs. LFADS, with and without sh).  Thus, while spike history couplings are needed for realistic spike train generation (as neural data contains dynamics that are not shared across the population, and thus cannot be captured by a low-d latent state), the performance benefits of LDNS are not due to spike history couplings alone.
>
> Your suggestion also allowed us to show that our post-hoc fitting of spike-history couplings provides a way to increase the realism of generated spike data not just for LDNS but for a class of VAE-based methods.
>
> **Table 1**
>
> |Method|$\mathbf{D_{KL},\text{psch}}$|RMSE pairwise corr|RMSE mean isi|RMSE std isi|
> |--|--|--|--|--|
> |AutoLFADS|$0.0040\pm2.2\times10^{-4}$|$0.0026\pm1.25\times10^{-5}$|$0.039\pm0.003$|$0.029\pm0.001$|
> |LDNS|$0.0039\pm3.9\times 10^{-4}$|$\mathbf{0.0025\pm1.1\times10^{-5}}$|$0.037\pm0.001$|$\mathbf{0.023\pm0.001}$|
> | | | | | |
> |AutoLFADSsh|$0.0036\pm2.1\times10^{-4} $|$0.0026\pm1.8\times10^{-5}$|$0.034\pm0.002$|$\mathbf{0.023\pm0.0001}$|
> |LDNSsh|$\mathbf{0.0016\pm6.2\times10^{-4}}$|$\mathbf{0.0025\pm1.07\times10^{-5}}$|$\mathbf{0.024\pm0.002}$|$\mathbf{0.023\pm0.001}$|
>
> **Post-hoc training for spike history, and choice of softplus vs. exponential**
> Taking rate predictions from LDNS (or any model, see previous paragraph), we optimize the spike history parameters with respect to the ground-truth spiking data. As a result, this alternative observation model does not impact the latents inferred by the S4-autoencoder. It allows us to independently improve generated spike trains, in particular to capture realistic autocorrelations, an important component towards accurate modeling of spiking data that is missing in the deep LVM literature as a whole. Furthermore, this opens the possibility for replacing our current version of spike history dependence with more sophisticated observation models without re-incorporating it into autoencoder or diffusion model training.
>
> We fit spike history dependence post-hoc since jointly optimizing with the autoencoder would likely interfere with the coordinated dropout regularization (Keshtkaran et al. 2019, see methods), which has been shown to be critical for other LVMs for neural dynamics. As we show in the above experiment, this post-hoc fitting process can be straightforwardly adapted to improve other models as well.
>
> Empirically, we found that using the exponential function was less numerically stable and resulted in higher loss values and poorer data fit than using the softplus-approximation. Softplus resulted in faster training convergence and a lower final loss (**Fig. R5**).   Furthermore, we believe that in the low spike count regime this is a fair approximation, as 99.8% of the inferred rates (across time bins) were less than 0.2 (per bin).
>
> **Other questions**
> **S4 vs. Mamba as an architectural choice**: S4 is parameterized as a time-invariant (stationary/autonomous dynamics) system, while Mamba and other context-selective models are parameterized as (and may be better suited for) non-stationary, input-driven dynamics, such as the human speech data. Both S4 and Mamba allow for time-parallelized training and length generalization. Exploring Mamba and other non-stationary linear recurrence models is an interesting idea of future work, but would go beyond the scope of this project.
>
> **Overestimation of correlation in attempted speech data**: Based on the reviewer’s suggestion, we increased the dimensionality of the latent space in LDNS from 32 to 48 and observed that the overestimation remains. We also analyzed the autoencoder and diffusion model separately and find that the inflated correlations are a result of the AE, not the diffusion part (**Fig. R9**). We do not yet fully understand the exact source of this overestimation, which we only observed for this particular dataset, and agree that further characterization would be beneficial.
>
> **Conditional generation of speech data**: While this application would be a natural extension for conditional generative modeling on complex behavioral data, in this work we focus on developing the methodology to enable such exciting applications in the future. Decoding speech from spiking data is an active research area, and consequently, evaluating the correctness of sampled data and decoded speech would be beyond the scope.

---

> > ### Comment · Reviewer_y7Q4 · 2024-08-08
> >
> > Thank you for your thorough response and for running the additional experiments. I especially appreciate the new comparisons of the spike history filter with LFADS, and for the additional baselines and visualizations. Overall, I have decided to increase my score and recommend accepting this paper.
> >
> > Nonetheless, I still remain unconvinced that the spike-history filter should be fit post-hoc and find the need for the softplus approximation unsatisfactory. Perhaps this implies the model should use a softplus nonlinearity rather than exponential during the initial training phase. Alternatively, joint training may alleviate the stability issues.
> >
> > To fit the model jointly with the spike history filter, I suggest using alternative masking schemes -- see e.g. a very recent paper [1] with alternative options that should work with the spike history filter.
> >
> > [1] Towards a "universal translator" for neural dynamics at single-cell, single-spike resolution. Zhang et al., arXiv:2407.14668

---

> > > ### Author Response · Authors · 2024-08-09
> > >
> > > Thank you for the positive response, detailed engagement with our work, and further increasing the score.
> > >
> > > We really appreciate the thorough questions about the spike history term and agree it would be interesting to see the effect of different nonlinearities in future work and how joint training would interact with coordinated dropout and alternative regularization schemes. Thank you for pointing us toward this paper; combining such approaches is indeed very promising.

---

### Official Review · Reviewer_AuW4 · 2024-07-13

**Soundness:** 4
**Presentation:** 2
**Contribution:** 3
**Rating:** 7
**Confidence:** 4

**Summary:**

The authors here propose a new autoencoder style latent variable model for neuroscience which flexibly adapts to variable time-series using S4 encoders and decoders. They then train diffusion based models with option behavioral covariates to generate realistic neural spiking data. Additionally, they use a more flexible spiking likelihood with history-filters which more accurately capture within neuron statistics. They evaluate their model on a synthetic lorenz-attractor example as well as multiple real-work datasets.

**Strengths:**

I see three separate contributions of this paper --  1) The ability to handle time-varying inputs in VAE based neural models is rare, as the authors point out, yet important in many practical neuroscience applications.  2) Using diffusion-based methods to generate samples from the auto-encoder inferred latents, allowing for the generation of variable length, realistic spiking data on unseen behaviors. I think the results of figure 5 are particularly interesting in this regard. 3) the addition of a history-based poisson likelihood, which better captures individual spiking statistics. This is a somewhat superficial contribution, but is nonetheless important to many of the authors' reported improvements.

**Weaknesses:**

I think the authors could do a better job communicating the disparate contributions and capabilities of this model more clearly to the reader, especially in contrast to existing approaches.

For example, I am having a hard time understanding to what extent the spike-history likelihood is important for this model. Many of the reported results in figures 2 and 3 are on the statistics of the spiking. Are these figures using the likelihood with the spike-history filter? I am assuming they are not, as the spike-history filter is explicitly specified as the likelihood used for figure 4, but not in figures 2 and 3. If they are not used here, why not? Further, can the authors demonstrate that this spike-history filter leads to a different scientific result? I.e. are the conditioned latents any more accurate or do they provide any more insight if one likelihood is used compared to another. The evaluations in the supplement demonstrate that the spiking statistics is better captured with this more sophisticated likelihood, but further discussion as to why this is important for scientific insight would be appreciated.

Overall, the authors primarily use spiking statistics as their measure of accuracy in this model throughout the paper. While the ability of a model to capture single-level neural statistics is important, it seems to me that the core use this model would have to an experimental practitioner is to visualize latents in an informative way in trial-varying data, potentially conditioned on behavior. Because of this, it would be nice to see more evaluations of the latent space, and not focus as much on the spiking statistics. Similar to the point above, it would be interesting to take a couple existing autoencoding LVMs used in neuroscience (like LFADs and others) and augment them with the same history-based Poisson likelihood. Then we would be able to see more clearly the contributions of the other features of the model such as S4 and diffusion, which I think are quite interesting, and their roles in identifying interpretable scientific latents where other models fail or cannot do so, such as in the trial-varying case or conditioned on behavioral covariates.

Lastly, there are many models that use autoencoder based approaches other than LFADS that could be compared to here but are not. It is of course not possible to compare to all of them, but some more thorough evaluation and discussion alongside existing approaches I think would significantly improve the paper. See for example, *Poisson Interpretable VAE, **Targeted neural dynamical modeling, ***PSID and Duncker and Sahani 2018. LFADs alone, especially one without the same history-dependent observation likelihood, does not provide a complete picture of the capabilities of latent identification of this model in relationship to other other approaches.

*) "Learning identifiable and interpretable latent models of high-dimensional neural activity using pi-VAE"
**) "Targeted Neural Dynamical Modeling"
***) "Modeling behaviorally relevant neural dynamics enabled by preferential subspace identification"

**Questions:**

Are the results reported in figures 2 and 3 using the spike history filter or just Poisson likelihoods?

What is the 'kde' in figure 3d?

**Limitations:**

See above. Further discussion and potential evaluation of this model alongside other approaches, and the dissociating the particular role of the likelihood as compared to the other model features, especially concerning scientific purposes, would be helpful for this paper.

---

> ### Author Rebuttal · Authors · 2024-08-06
>
> We thank the reviewer for finding our work relevant, and our approach sound. Based on your suggestions, we have now performed new baseline experiments with additional VAE-based models. We also performed latent space analyses of sampled spikes from all models (using PCA and LFADS embeddings) to supplement the spike statistics. We additionally inspected the quality and interpretability of LDNS-derived latents as suggested. Lastly, we augmented LFADS with spike history dependence for a more clear comparison, and clarify how and why we account for it post-hoc.
>
> **In summary, LDNS remains the most performant model overall for spike generation, and its single-trial latents are informative of behavior**. We detail the experiments and analyses below, and clarify the disparate contributions and capabilities of LDNS more clearly, especially in contrast to existing approaches. We hope these additions address the reviewer’s concerns and enable them to raise their score.
>
> New figures (**Fig. Rx**) are in the **PDF**.
>
> **New VAE-based baseline experiments**
> We implemented two new VAE-based models suggested by the reviewer, TNDM (Hurwitz et. al. 2021\) and pi-VAE (Zhou et al, 2020), on the unconditional monkey reach task, and find that LDNS is superior in realistic spike generation (**Fig. R1, see PDF**).
>
> **TNDM**: We trained TNDM on the monkey dataset using the original proposed architecture and hyperparameters. We used 5+5 latent factors, the maximum in the original paper, and used the prior N(0,1) to sample the RNN initial states for unconditional generation.
> **pi-VAE**: We trained pi-VAE using the original architecture (and model hyperparameters), which does not consider temporal dynamics and treats each time point as an independent sample. Thus, pi-VAE samples exhibit poor spiking statistics and no temporal structure (**Fig. R1, R7,** yellow).
>
> **Evaluation of LDNS latents**
> We agree that, while generation of realistic spike trains in terms of spike statistics is important, many neuroscientists are interested in the extracted latents. Thus, we analyzed LDNS-extracted latents of unseen test samples from the monkey reach dataset and colored them by reach angle (**Fig. R6,** straight reaches only for visualization). Note that behaviorally relevant information is clearly reflected in single-trial latents of LDNS (**Fig. R6**, bottom row). Furthermore, this latent structure is preserved when sampling from LDNS conditioned on unseen velocity trajectories (**Fig. R6**, top row). We will add this result to Fig. 5, highlighting the ability of LDNS to extract meaningful latents.
>
> **Contribution of spike history and extension to LFADS**
> Spike trains exhibit autoregressive temporal dependencies not shared across the population, and therefore spike-history coupling is needed for any low-dimensional model to achieve realistic spike train generation.
> The spike history model in LDNS is fit post-hoc after autoencoder training, and therefore does not impact the inferred latents. This allows us to independently improve LDNS-generated spike trains and their autocorrelations, an important component towards accurate modeling of spiking data. Directly optimizing the spike history features during autoencoder training might interfere with the coordinated dropout regularization (Keshtkaran et al. 2019, see methods), which has been shown to be critical for other LVMs of neural dynamics.
> Lastly, we view the post-hoc augmentation of the observation model with spike history as a key modular contribution, which can be flexibly applied to other generative models. Based on the reviewer’s suggestion, we have additionally extended LFADS with spike history dependence, which improves its performance (**Fig. R1,4**). However, LDNS with spike history is still superior or on par on spike metrics against this extended version of LFADS. We also emphasize that, both with and without spike history, LDNS outperforms its LFADS counterpart, and we highlight the corresponding comparisons in Table 1 (see Reviewer y7Q4). We will include these results and clarifications in the revised paper.
>
> **Contributions of LDNS (in contrast to other approaches):**
>
> - LDNS combines S4 and diffusion models for the purpose of accurately modeling and generating neural spiking data—a task often ignored by other LVMs designed for neural data analysis (such as LFADS, pi-VAE, and TNDM).
> - The S4 autoencoder and diffusion model are trained in separate stages, offering modularity and easier debugging, while both components naturally account for temporal dependencies (unlike pi-VAE).
> - Furthermore, S4 is autoregressive, similar to other RNN-based models, but empirically we found it to perform better when extending past the training trial length and more readily handle variable length data (compared to LFADS), due to the masked training procedure we propose here.
> - Finally, the spike history-dependent observation model is modular and can be optimized post-hoc using rate predictions of any model, offering flexibility while improving spike generation quality.
> - One feature provided by other neural-behavioral analysis models (such as pi-VAE and TNDM) is an explicit disentangling of neural vs. behavior-relevant latents, which we did not consider but is a possible future extension for LNDS.
>
> **Are the results reported in figures 2 and 3 using the spike history filter or just Poisson likelihoods?** For the results in Figs. 2, 3, 4(b,c,d), 5, we only use the Poisson likelihood with no history terms. In the  Lorenz experiment, we know the ground-truth generative model uses Poisson emission, and for the human BCI dataset we omitted them due to the large bin size (20ms) used here.
>
> **What is the 'kde' in figure 3d?**  A Kernel Density Estimate (KDE) with a Gaussian kernel to estimate the population spike count distribution. We will clarify both points in the revision.

---

> ### Comment · Reviewer_AuW4 · 2024-08-07
>
> This is a very thorough and impressive response. I think these additional figures paint a much more complete picture of the model's capabilities in a way that now clarifies it's disparate contributions compared to competing approaches. I believe these will substantially increase the work's impact.
>
> Particularly, R2 and R3 clearly demonstrate the LDNS addresses something that is lacking in modern generative LVMs for neural data, and I believe they should be highlighted in the main manuscript. R1 also adds important baselines that would be nice to prominently highlight as well (particularly including the LFADS sh).  The additional latent space characterizations (R6-R8) helps future model practitioners get an idea of the model's scientific utility. It would be nice if possible to include some of this in the main body of the paper, but I don't believe it is necessary.
>
> I have updated my score accordingly and am excited to hopefully see this work at this year's NeurIPs.

---

> > ### Author Response · Authors · 2024-08-09
> >
> > Thank you so much for the prompt and kind response, and now recommending acceptance.
> >
> > We appreciate your thorough engagement with our work and detailed comments, which allowed us to clarify our contributions. We agree that adding these analyses will strengthen the paper.

---

### Author Rebuttal · Authors · 2024-08-06

We thank all the reviewers for their constructive and detailed engagement with our work, resulting in many helpful comments and opportunities for clarification. We are especially grateful for several reviewers’ assessments that the work is “original” (mYEj), well written and clearly motivated (y7Q4, mYEj, pMhd), technically sound (mYEj, AuW4) with “impressive performance” (y7Q4), and represents a significant contribution to the field (pMhd, y7Q4).

We agree with many of the suggestions and questions raised, and respond individually to each review in detail. We summarize here the main points and new experiments. We hope that our response addresses their questions and concerns, allowing all reviewers to recommend our work for acceptance.

**Summary of original contribution**
LDNS combines S4-based autoencoders and diffusion models for both extracting low-dimensional latent dynamics from neural population spiking data and generating realistic spike trains, an aspect of evaluation often ignored by other works in the literature. On a commonly used monkey reach task, LDNS outperforms or matches LFADS in all metrics. By equipping LDNS with a modular spike history-dependent observation model, it surpasses LFADS on all metrics and accurately captures spike autocorrelation. Moreover, LDNS works well with variable trial lengths and shows length generalization abilities. Finally, LDNS can conditionally generate realistic spiking data based on behavioral covariates for the monkey reach task.

**Summary of new results**
In response to the reviewers, we substantially expanded our evaluation with new experiments: On the monkey reach task, LDNS outperforms two newly suggested baseline models **(Fig. R1, see PDF**). We also augment LFADS with spike history and report increased performance, but it is still surpassed or matched by LDNS-sh on the same task (**Fig. R4, Table 1 in response to pMhd**). In addition, we evaluate LFADS on the other two datasets and find that unlike LDNS, LFADS failed at length generalization (**Fig. R2,3**). Moreover, we show that on the monkey task, LDNS latents contain behaviorally relevant information, suggesting their utility in analyzing and visualizing neural data (**Fig. R6**). Finally, we evaluate the dynamics of generated spikes sampled from LDNS (and other models) via PCA and LFADS embedding, and find that LDNS accurately captures the dynamics of real data (**Fig. R7,8**).

**1\. New baseline experiments (AuW4, pMhd, mYEj)**
We conducted numerous additional baseline experiments to demonstrate the contributions of LDNS relative to existing (VAE-based) methods:
We implemented and fit two other methods, pi-VAE (Zhou and Wei, 2020\) and TNDM (Hurwitz et al. 2021), to the monkey dataset (**Fig. R1**), and find that LDNS still consistently outperforms the others on spike generation. Details in individual responses to AuW4 and PMhd.
We added our proposed spike history observation model (see point 3 below) to the LFADS baseline on the same task, which increases performance but still not to the level of LDNS with spike history. We note that LDNS without spike history is still superior or on par compared to all baselines, highlighting the contribution from the latent diffusion model.

**2\. Evaluating LFADS on the other datasets (mYEj)**
We further applied LFADS on the Lorenz and human dataset, in particular, to assess the length-generalization ability of LDNS relative to existing models of neural dynamics.
It is well established that LFADS fits the Lorenz dataset well (Pandarinath et al. 2018). However, LFADS struggles with length generalization (**Fig. R2**), while LDNS samples can be generated at 16x the training trial length (Fig. 2c), a key feature made possible by S4.
We also fit LFADS to spike recordings from the human brain, a challenging task due to highly heterogeneous trials with variable length, which LDNS can handle thanks to its architecture and masking scheme. To fit LFADS, we cut the data into equal length segments. Despite successful training, LFADS failed to capture dynamics beyond the cut-off during sampling (**Fig. R3** vs. Fig. 3). This highlights the capability of LDNS to deal with variable-length data, which is highly relevant for many neuroscience datasets (as pointed out by AuW4). Further discussion in response to reviewer mYEj.

**3\. Contribution of, and clarifications on, spike-history coupling (AuW4, y7Q4)**
We clarify that the spike-history observation model is applied **after** first training the autoencoder model with standard Poisson likelihood. The history model increases the quality of the sampled spikes, and can be flexibly applied to other models. Addressing questions raised by reviewers, we ran a new experiment augmenting LFADS with spike history, which increases the realism of generated samples from this baseline model (**Fig. R4**), though it still underperforms LDNS with spike history.

**4\. Analysis of LDNS latents (AuW4, mYEj)**
We agree that single-trial latent analysis is of interest to the neuroscience community, and have therefore evaluated the quality of LDNS latents (**Fig. R6**). On the monkey reach dataset, we plot PCs of the inferred latents and show that behaviorally relevant information (colored by reach angle) is reflected in the latent space of LDNS, a desirable property for neuroscientists aiming to visualize their data.

**5\. Evaluating dynamics of LDNS samples (mYEj)**
We evaluated the realism of LDNS samples in terms of underlying low dimensional dynamics using PCA (applied to smoothed, generated spikes vs. true data, **Fig. R7**). This suggestion clearly distinguished failure cases (e.g., no temporal dependencies in samples generated using pi-VAE). As suggested by reviewer mYEj, we also embedded the generated spikes using LFADS, and show that LFADS latents extracted from LDNS spikes more closely capture the broad distributions of the true data than, e.g., spikes sampled from LFADS (**Fig. R8**).

---

### Decision · Program_Chairs · 2024-09-25

**Decision:**

Accept (spotlight)

**Comment:**

There was broad consensus among the reviewers that this paper is technically solid and interesting for the NeurIPS community. The additional empirical validation done in the authors' rebuttal was found to be particularly useful.